# Oxygen suppression of macroscopic multicellularity

G. Ozan Bozdag [1✉], Eric Libby [2,3,4], Rozenn Pineau[1,5], Christopher T. Reinhard [6,7] & William C. Ratcliff [1,8✉]

Atmospheric oxygen is thought to have played a vital role in the evolution of large, complex multicellular organisms. Challenging the prevailing theory, we show that the transition from an anaerobic to an aerobic world can strongly suppress the evolution of macroscopic multicellularity. Here we select for increased size in multicellular 'snowflake' yeast across a range of metabolically-available $O_2$ levels. While yeast under anaerobic and high-$O_2$ conditions evolved to be considerably larger, intermediate $O_2$ constrained the evolution of large size. Through sequencing and synthetic strain construction, we confirm that this is due to $O_2$-mediated divergent selection acting on organism size. We show via mathematical modeling that our results stem from nearly universal evolutionary and biophysical trade-offs, and thus should apply broadly. These results highlight the fact that oxygen is a double-edged sword: while it provides significant metabolic advantages, selection for efficient use of this resource may paradoxically suppress the evolution of macroscopic multicellular organisms.

[1] School of Biological Sciences, Georgia Institute of Technology, Atlanta, GA, USA. [2] Integrated Science Lab, Umeå University, Umeå, Sweden. [3] Department of Mathematics and Mathematical Statistics, Umeå University, Umeå, Sweden. [4] Santa Fe Institute, Santa Fe, NM, USA. [5] Interdisciplinary Graduate Program in Quantitative Biosciences, Georgia Institute of Technology, Georgia, USA. [6] School of Earth and Atmospheric Sciences, Georgia Institute of Technology, Atlanta, GA, USA. [7] NASA Astrobiology Institute, Alternative Earths Team, Riverside, CA, USA. [8] NASA Astrobiology Institute, Reliving the Past Team, Atlanta, GA, USA. ✉email: ozan.bozdag@gmail.com; william.ratcliff@biology.gatech.edu

While simple multicellularity first evolved in prokaryotes as early as 2.6 billion years ago[1,2], these lineages never evolved to be large and complex[3,4] (i.e., exhibiting extensive cellular differentiation with numerous cell types). This is in striking contrast to eukaryotic life, which independently evolved an impressive array of large, complex multicellular forms[5–7]. Yet even within eukaryotes, it was not until several hundred million years after the diversification of their crown groups that macroscopic multicellular linages arose and underwent a global evolutionary radiation[8–11]. During this prolonged lag (1.6–0.8 Ga), the concentration of $O_2$ in Earth's ocean-atmosphere system was much lower than today (perhaps below ~1% of the present atmospheric level, PAL)[12,13]. By the start of the Ediacaran (635 Ma), millimeter-to-centimeter scaled macroscopic multicellular eukaryotes had evolved and spread across the oceans[11,14–19]. These organisms, which represent the first globally distributed forms of macroscopic multicellularity, had a thin tissue layer and were diffusion-limited due to the absence of sophisticated cellular differentiation[15,20,21], with key innovations such as circulatory systems evolving later in metazoans. The abrupt radiation of macroscopic multicellularity in the Ediacaran oceans coincides with evidence for a global rise in ocean-atmosphere oxygen levels (~800–550 Ma)[12,22–24]. Although this geobiological observation is largely undisputed, the role of oxygen in the evolution of macroscopic multicellularity remains a topic of intense debate[25–29].

The oxygen control hypothesis (OCH) is the dominant explanation causally linking the concentration of oxygen with multicellular size[27]. The OCH posits that, in organisms that lack a circulatory system, the ability for oxygen to diffuse into an organism places limits on organism size. The OCH thus predicts that increasing atmospheric $pO_2$ should generally increase the depth to which $O_2$ can diffuse, monotonically increasing the maximum size that can be attained before diffusive $O_2$ limitation impedes growth[27]. The central prediction of the OCH, that increased atmospheric $pO_2$ can support larger multicellular organisms, has largely been supported by mathematical models[30–34], comparative work in natural environments[35,36], and manipulative experiments with modern animals[37], most of which show that low $pO_2$ constrains body size (but see ref. [38]).

The OCH, however, is not a general model for oxygen-size relationships, and it cannot be applied to the question of how the transition from an anaerobic to a microaerobic world would have affected organismal size. This is because the OCH assumes that organisms are obligately aerobic, a derived metabolic strategy suited to an already well-oxygenated world. Stem-group eukaryotes, for example, which evolved in the low-$O_2$ oceans of the Proterozoic[39], were mixotrophic[40–43], and mixotrophy remains common today in oxygen-minimum zones[44–46]. Further, the OCH has focused almost exclusively on physiological rather than evolutionary timescales, examining the immediate physiological impacts of $O_2$ but ignoring evolutionary feedbacks that are ultimately responsible for systematic changes in multicellular size. Oxygen is a valuable resource, both increasing the efficiency of metabolism up to 16-fold[47] and stochiometrically unlocking the metabolic potential available in non-fermentable carbon[48]. Thus, the evolution of multicellular size should be viewed through the lens of evolutionary trade-offs (i.e., multicellular organisms can potentially gain a benefit by being large, but may suffer increased diffusive constraints on access to $O_2$ as a result, reducing the metabolic benefits of using oxygen), rather than through the strictly physiological lens of the OCH.

To our knowledge, no prior work has examined the relationship between oxygen availability and size over evolutionary timescales in a diffusion-limited multicellular organism, a gap that is partly due to the lack of suitable model systems. Here, we examine the effect of oxygen on multicellular size using a combination of experimental evolution, synthetic biology, and mathematical modeling, using yeast model system of undifferentiated multicellularity. First, we perform an ~800 generation selection experiment, examining the ability of snowflake yeast to evolve larger size under a range of $O_2$ levels. While large size readily evolves in anaerobic and high-$O_2$ conditions (near modern levels), it is suppressed at intermediate $pO_2$. To confirm that the results of our evolution experiment reflect selection acting on multicellular size, and are not confounded by metabolic differences or parallel evolutionary changes, we genetically engineer small and large snowflake yeast and examine their fitness under varying $pO_2$. Finally, we recapitulate our results in a simple evolutionary model, highlighting how selection for efficient use of oxygen when it is limiting can strongly constrain the evolution of increased multicellular size.

Taken together, this work suggests that the oxygenation of Earth's oceans was neither necessary nor sufficient for the evolution of large multicellular size. In fact, when limiting to growth, oxygen may strongly suppress the evolution of large size by favoring smaller organisms that can better utilize it. This result has far-reaching implications for the origin of complex life, suggesting that the transition from an anaerobic to microaerobic world may have acted as a powerful constraint on the evolution of large, diffusion-limited organisms, providing new insight into the mechanisms underlying the so-called "boring billion" years of Earth's evolutionary history between ~1.8–0.8 Ga[49].

## Results

**Model system.** We constructed our initial snowflake yeast by deleting the *ACE2* open reading frame in the unicellular strain Y55. This leads to incomplete separation of mother and daughter cells, resulting in the formation of multicellular clusters. Snowflake yeast possess an emergent multicellular life cycle in which clusters grow until packing strain generated by cellular division causes cell–cell fracture, giving rise to new snowflake yeast clusters[50,51]. Mutations are efficiently segregated between groups, and individual clusters are primarily monoclonal[52,53]. Snowflake yeast populations undergo ~5 generations of growth per day, readily adapting to size-based selection by evolving to be larger[51,54].

Snowflake yeast grow as approximately spherical clusters of densely packed cells, but little is known about the extent to which interior cells are limited by access to oxygen—a constraint that is necessary for their use as a model system of diffusion-limited multicellularity. We examined the diffusion depth of oxygen within snowflake yeast by genomically integrating the MitoLoc construct (*preSU9-GFP* + *preCOX4-mCherry*)[55] into our ancestral snowflake yeast strain, allowing us to visualize mitochondrial activity throughout the cluster. The preSU9-GFP marker localizes to the F0-ATPase subunit-9 independent of aerobic respiration, whereas preCOX4-mCherry localizes to the mitochondria only when there is an active proton gradient in the organelle. Only cells near the surface were capable of respiration, while the rest of the cluster interior is effectively anaerobic (Fig. 1a). Furthermore, while an average of 28% of cells/cluster showed aerobic activity under standard batch culture conditions, this increased to 56% with oxygen supplementation (Fig. 1b; Mann–Whitney $U = 262$, $n_1 = 29$, $n_2 = 37$, $p = 0.0003$, two-tailed). Despite their branched growth form, respiration in snowflake yeast is strongly diffusion limited.

**Examining the impact of $pO_2$ on size via experimental evolution.** *Saccharomyces cerevisiae* has evolved a somewhat unusual form of mixotrophy: it ferments glucose, even in the presence of

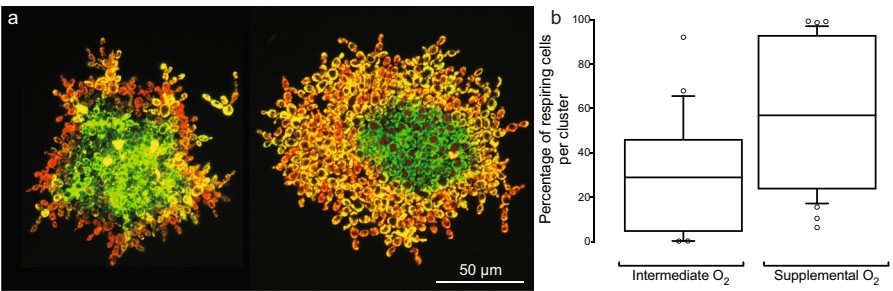

**Fig. 1 Interior cells are O$_2$ limited and largely incapable of respiration. a** Images of representative snowflake yeast clusters expressing the MitoLoc[55] construct (left: under intermediate O$_2$, right: under supplemental O$_2$). Peripheral cells are actively respiring, as shown by the dual staining of preCOX4-mCherry, which only enters mitochondria with an active proton gradient, as well as pre-SU9-GFP, which labels all mitochondria. In contrast, few internal cells are respiring. **b** Supplementing our batch culture with additional oxygen doubled the fraction of respiring cells per cluster, from an average of 28% to an average of 56% (Mann–Whitney test, $U = 262$, $p = 0.0003$, two-tailed; $n = 29$ and 37 for the intermediate and supplemental O$_2$ treatments, respectively). The whiskers are drawn down to the 10th and up to the 90th percentiles, and data outside of the whiskers are shown as individual points. The lines in the middle of the boxes show the median values. Source data are provided as a Source Data file.

oxygen (a process known as the Crabtree effect)[56]. To directly relate oxygen availability to O$_2$-mediated growth, we grew our yeast on a non-fermentable carbon source, glycerol, forcing it to respire throughout the culture cycle. We evolved snowflake yeast under three different metabolically-available pO$_2$ regimes: (1) we generated mutants incapable of respiration (known as "petite" yeast) which grow anaerobically even in the presence of atmospheric oxygen; (2) we grew strictly aerobic yeast under intermediate O$_2$ (~24% Present Atmospheric Level [PAL]) and (3) high O$_2$ (72% PAL). Finally, as a control for the effects of modifying yeast metabolism, we also evolve mixotrophic yeast under intermediate oxygen (~25% PAL O$_2$). We evolved five replicate populations of each treatment (20 populations total) for 145 daily transfers, or ~812 generations (generations/day measured for all populations at 0, 50, 100, and 145 transfers). Each day, the yeast experienced 24 h of selection for faster growth followed by a single round of selection for larger multicellular size (4-min settling selection[54], see "Methods" section for details).

Surprisingly, greater oxygen availability did not necessarily foster the evolution of larger size. The two treatments that evolved to be the largest were aerobic respiration under high O$_2$ (97% increase in mean size; $p < 0.0001$, $F_{3,24} = 12.19$, Dunnett's test in one-way ANOVA) and anaerobic fermentation (93% increase in mean size; Fig. 2a and Supplementary Figs. 2–5; $p < 0.0001$, $F_{3,22} = 256.7$, Dunnett's test in one-way ANOVA). However, strictly aerobic yeast grown under intermediate O$_2$ did not evolve to be significantly larger, even after 145 days of daily size selection (8.9% increase, $p = 0.1$, $F_{3,24} = 1.734$, Dunnett's test in one-way ANOVA), while the mixotrophic control cultured under intermediate O$_2$ evolved to be 37% larger than their ancestor ($p < 0.0001$, $F_{3,24} = 329.1$, Dunnett's test in one-way ANOVA). Indeed, rather than a positive, monotonic relationship, such as that described by the OCH, we find that intermediate oxygen availability strongly suppressed the evolution of larger size relative to either anaerobic or highly aerobic conditions after 145 days of evolution in a total of 20 evolved populations (Fig. 2c; $p < 0.0001$, Welch's $t$-test, $t = 8.6$, df $= 18$).

Prior work has demonstrated that snowflake yeast evolve to be larger primarily by increasing their cellular aspect ratio. This heritable, cell-level trait underlies the emergence of greater multicellular size by reducing the strain on cell–cell connections caused by cellular packing[51]. To determine if yeast cultured under both low and high O$_2$ were employing similar biophysical mechanisms to increase size, we examined the relationship between mean cellular aspect ratio and mean cluster size for our three ancestors and 20 evolved populations after 145 days of

evolution (Fig. 2d). Across all 23 populations, cellular aspect ratio explained 92% of the variation in multicellular size (Fig. 2d; $y = 36.44x - 11.97$, $p < 0.0001$). The strong linear correlation suggests that the mechanistic underpinnings of increased multicellular size (increased packing efficiency through cellular elongation) was similar in all experimental populations, and was unaffected by environmental or metabolic differences.

**Testing O$_2$-mediated size suppression via synthetic construction.** Experimental evolution is fundamentally multi-dimensional, with long-term evolutionary change being affected by genetic historical contingencies[57]. To rigorously test the hypothesis that selection favors large size under both anaerobic and highly aerobic conditions, but favors small size when O$_2$ is limiting, we constructed otherwise isogenic large and small snowflake yeast. To find a short list of candidate mutations for constructing a large-sized phenotype, we sequenced the genome of a large snowflake yeast, and identified 36 de novo mutations (see Supplementary Data 1 for a list of mutations). We then screened seven single-gene deletions for their effect on size, and found that the combined deletion of GIN4 and ARP5 resulted in the formation of more elongated cells, which in turn increased the average radius of multicellular groups by 69.7% in petite (mitochondria incapable of oxidative respiration) and 29.3% in grande (functional mitochondria) snowflake yeast (Fig. 3a; $p < 0.0001$, $F_{3,10759} = 1398$, Sidak's Method in one-way ANOVA). These mutations increase group size by modifying the biophysics of snowflake yeast growth. Specifically, Δarp5 increases cellular aspect ratio, decreasing the packing fraction within clusters and increasing their biophysical toughness[51], and Δgin4 increases the size of bud scars, potentially increasing the strength of cell–cell connections.

We competed our synthetic strains under the same conditions as our evolution experiment. Specifically, we competed small vs. large snowflake yeast phenotypes under three different pO$_2$ levels and four metabolic conditions: (1) anaerobic (petite) snowflake yeast under 0% PAL metabolically-available O$_2$; strictly aerobic (grande) snowflake yeast under both (2) intermediate-O$_2$ and (3) high-O$_2$ conditions (25% PAL and 75% PAL, respectively); and (4) mixotrophic (grande) snowflake yeast under intermediate-O$_2$ (25% PAL), as a control. For each competition, we calculated the daily selection coefficient (proportional change in the frequency of individual clusters) of large vs. small snowflake yeast, $r_w$, after 24 h of growth and one round of settling selection for two consecutive days of growth and size selection. Consistent with the outcome of our experimental evolution (Fig. 2), engineered large snowflake yeast were strongly favored by

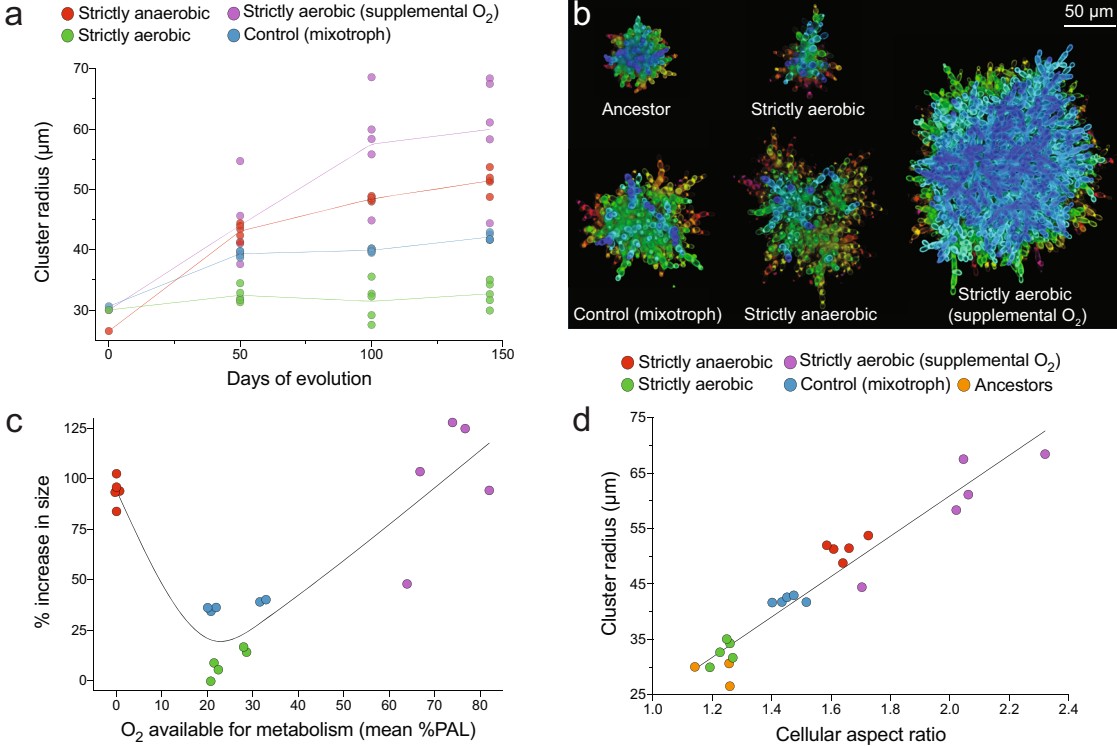

**Fig. 2 The evolution of large size in snowflake yeast is constrained under low-oxygen conditions. a** Temporal dynamics of size evolution in each treatment. **b** Confocal images of representative clusters for each treatment after 145 transfers. Color indicates z-axis depth. **c** Low oxygen constrained the evolution of large size, relative to anaerobic or highly aerobic conditions. Shown here are the final mean size of snowflake yeast clusters within each of the 20 populations shown in **a**, plotted against the average metabolically-available pO$_2$ from each experimental microcosm. Line (**c**) is a spline with four knots. **d** Larger multicellular size evolved through increased cellular aspect ratio in all treatments. The aspect ratio and cluster size values are from the three ancestral genotypes (strictly aerobic, strictly anaerobic, and mixotrophic control) and 20 evolved populations. Source data are provided as a Source Data file.

selection under anaerobic (mean $r_w = 0.26$, $p < 0.0001$) and highly aerobic (mean $r_w = 0.24$, $p < 0.0001$) conditions, while small snowflake yeast were favored under intermediate O$_2$ (Fig. 3b; strictly aerobic mean $r_w = -0.3$, $p < 0.0001$; mixotrophic control mean $r_w = -0.28$, $p < 0.0001$; $F_{3,26} = 265.1$, Tukey's HSD; one-way ANOVA).

Both *gin4Δ* and *arp5Δ* mutations have pleiotropic effects, reducing unicellular growth rates in a similar manner under both intermediate and high-O$_2$ conditions. Unicellular *gin4Δ* mutants had a relative fitness of 0.898 and 0.900 under intermediate-O$_2$ and supplemental-O$_2$ conditions, respectively. Similarly, unicellular *arp5Δ* had a relative fitness of 0.937 and 0.949 under intermediate-O$_2$ and supplemental-O$_2$ conditions, respectively. This reduction in growth rates should not affect our interpretation of the engineered multicellular-strain competition experiment: despite this growth cost, the engineered strain that forms larger clusters still outcompeted the smaller competitor under both anaerobic and high-O$_2$ conditions (but not under intermediate-O$_2$).

**General evolutionary model of O$_2$-size relationships.** To contextualize our experimental results and provide general insight into the role of oxygen in the evolution of organismal size, we developed a first-principles evolutionary model. This model is not intended to directly recapitulate our experimental results above, but instead is designed to examine how oxygen affects the evolution of multicellular size more generally. We consider a simple, spherical, diffusion-limited multicellular organism under size selection across a range of oxygen environments (0–100% PAL). We assume that cells within the organism with access to oxygen

respire aerobically, while cells that are anoxic ferment, and then calculate the size that maximizes fitness as a function of environmental pO$_2$.

We calculate fitness ($w$) as the expected number of offspring over time ($t$) as a function of organism size ($s$):

$$w(s, t) = (2p(s))^{t/\tau(O,s)}, \qquad (1)$$

where $p(s)$ is the probability that an organism of size $s$ survives to double in size, and thus reproduce, and $\tau(O,s)$ is the time an organism of size $s$ takes to double in volume given a certain pO$_2$. As with most models of early multicellularity, we assume that there is a fitness-related advantage associated with increased size and link survival to organism size with the expression $p(s) = 1 - 0.5e^{-ks}$, where $k$ is a positive scaling parameter (with smaller values of $k$, organisms need to be larger to obtain the same survival benefit of increased size, see Supplementary Fig. 6). The range of $p(s)$ ensures a population can grow, regardless of organism size, though larger size improves survival with diminishing returns.

To determine the time to reproduction $\tau(O,s)$, we assume that cells grow exponentially such that the rate of change in organism size is given by $ds/dt = \lambda_a(s)s$, where $\lambda_a(s)$ is the average cell growth rate for an organism of size $s$. Aerobic growth quickly depletes oxygen within a group of cells, resulting in an anaerobic core[58,59]. For each O$_2$ environment (0–100% PAL), we calculate the proportion of anaerobic cells within the spherical organism, $p_f = (1 - \theta/r)^3$, where $r$ is its radius and $\theta$ is the distance from the surface with enough oxygen to sustain respiration. Based on prior experiments, we assume that respiration yields 6.8 times as

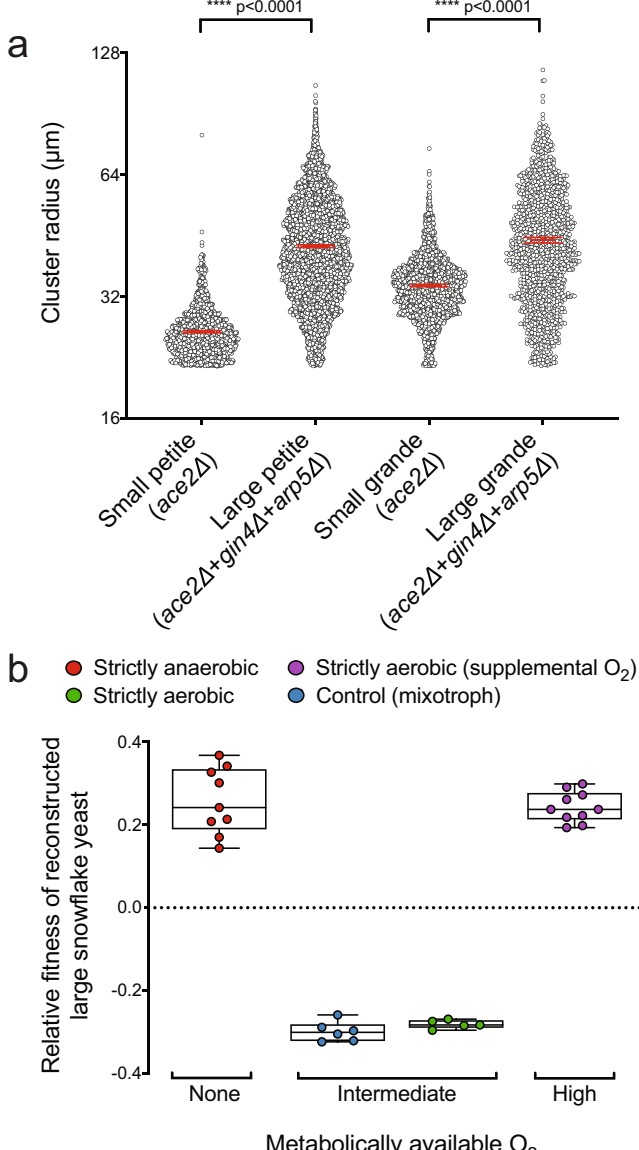

As expected, in an anaerobic environment where all organisms have the same metabolic output, larger organisms have an advantage due to their greater survival (Fig. 4b, blue line). However, when $O_2$ is present but cannot diffuse very deeply into tissue, smaller organisms have greater Darwinian fitness because the increased growth rate gained by aerobic respiration more than compensates for their lower survival (Fig. 4b, purple line). As oxygen becomes more abundant in the environment larger organisms become the most fit, because deeper oxygen diffusion mitigates the trade-off between growth rate and size (Fig. 4b, red line). The key qualitative result of this model—in which the transition from an anaerobic to a weakly aerobic environment strongly suppresses the evolution of large size (Fig. 4c)—is quite robust. Oxygen suppresses the evolution of larger size when respiration provides as little as 0.1% more growth than fermentation (Supplementary Fig. 7a). Notably, this effect is even more pronounced when we compare obligately aerobic to obligately anaerobic organisms (Supplementary Fig. 8), rather than the mixotrophic organisms of the base model.

The crucial variable that determines how selection acts on size is $\theta$, the depth to which $O_2$ diffuses within the organism (Fig. 4c). For spherical organisms, we can directly relate $\theta$ to $pO_2$ with the expression $\theta = \left(\frac{6D_e S_o}{\varphi}\right)^{0.5}$, where $D_e$ is the diffusion coefficient of oxygen through the organism (base value $= 1.25 \times 10^{-5}$ cm$^2$ s$^{-1}$ [54]), $S_o$ is the concentration of oxygen in the environment (8.24 mg l$^{-1}$ at 25 °C and 100% $pO_2$, scaled for lower partial pressures using Henry's law), and $\varphi$ is the volumetric reaction rate of oxygen within the organism (base value $= 46$ mg s$^{-1}$ l$^{-1}$)[54]. Oxygen suppression of large size occurs between 0.0001 and 1% $pO_2$ PAL, even when we consider large ($\geq$100-fold) variation in in the rates of oxygen diffusion or consumption (Fig. 4d and Supplementary Fig. 9) and 10,000-fold variation in size selection strength parameter $k$ (Fig. 4d and Supplementary Fig. 7b).

## Discussion

We observe a striking agreement between the predictions of our simple theoretical model and the outcome of an ~800 generation directed evolution experiment, in which larger multicellular organisms readily evolved when they could not use oxygen or when oxygen was plentiful, but were inhibited from evolving larger size at intermediate $pO_2$. We show that selection was acting directly on the fitness consequences of size in our evolution experiment, and that all 20 of our experimental populations utilized a similar biophysical mechanism for evolving larger size. Taken together, our work demonstrates that when limiting, oxygen can strongly suppress the evolution of increased size in a simple, diffusion-limited multicellular organism.

Previous work on the role of oxygen in the evolution of organismal size has argued that increasing atmospheric oxygen should monotonically increase the size to which organisms evolve[30,32,33]. Critically, these models predict little difference between anaerobic and microaerobic environments, focusing on the maximum size that is physiologically achievable at a given $O_2$ abundance[30-32]. Our approach differs in two key ways: first, we do not limit ourselves to obligate aerobic respiration, because fermentation/mixotrophy is taxonomically widespread[44,63], ancestral within eukaryotes[40,41,64], and of critical biological importance within low oxygen environments. Second, we take an evolutionary rather than physiological approach, which allows us to make testable predictions about how size should change under different oxygen regimes. Oxygen is a valuable (though generally non-rivalrous) resource, allowing organisms to generate significantly more energy and biomass through increased metabolic efficiency and the ability to respire non-fermentable substrates[48,65]. When $O_2$ is present but access is limited by

**Fig. 3 Testing $O_2$-mediated selection on size via synthetic construction. a** Engineered ($\Delta$ace2+$\Delta$gin4+$\Delta$arp5) snowflake yeast were 62.7% larger in a petite (mitochondria incapable of respiration) background ($n = 1920$ for small petite clusters and $n = 4987$ for large petite clusters; $p < 0.0001$, $F_{3,10759} = 1398$, Sidak's Method in one-way ANOVA), and 29.3% larger in a grande (functional mitochondria) background ($n = 2308$ for small grande clusters and $n = 1548$ for large grande clusters; $p < 0.0001$, $F_{3,10759} = 1398$, Sidak's Method in one-way ANOVA). Error bars show mean with 95% CI. **b** Engineered large snowflake yeast had higher relative fitness than small clusters when oxygen was not used for growth (anaerobic metabolism) or in a high-$O_2$ environment (~72% $pO_2$) (Tukey's HSD in one-way ANOVA, $p < 0.0001$, $F_{3,26} = 265.1$). When $O_2$ was more limiting, selection favored small-sized snowflake yeast. Relative fitness is reported as a daily change in proportion of the competing strains[89]. Central lines show median values, error bars show min-to-max values, and data points show the result of each independent fitness experiment, i.e., $n = 9$ for anaerobes, $n = 5$ for control mixotrophes, $n = 6$ strictly aerobic (Intermediate $O_2$), and $n = 10$ for strictly aerobic (High $O_2$). Source data are provided as a Source Data file.

much growth as fermentation[60-62] and calculate organismal growth rate (Fig. 4a) as the weighted average of cells growing via respiration ($\lambda_r$) and fermentation ($\lambda_f$) within the organism: $\lambda_a = p_f\lambda_f + (1 - p_f)\lambda_r$.

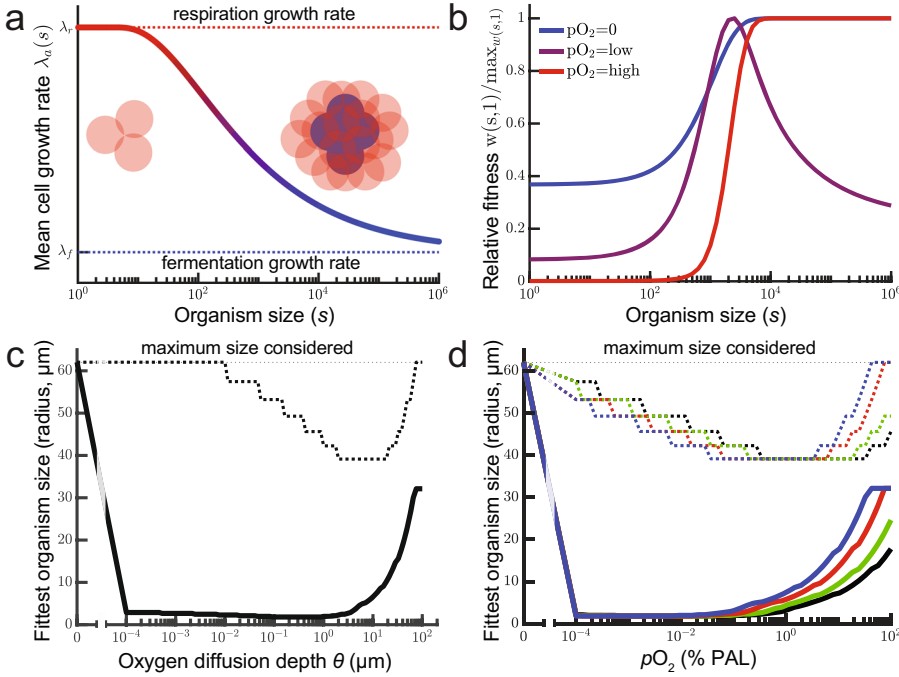

**Fig. 4 Trade-offs constrain the evolution of size under intermediate O₂ conditions.** We modeled the $O_2$-dependent evolution of size in a simple, diffusion-limited organism. **a** Larger organisms can develop an anaerobic core of cells (blue shading), decreasing their average growth rate. **b** Large size is adaptive when oxygen is absent or abundant (100% PAL), but is maladaptive when oxygen is present but cannot efficiently reach internal cells via diffusion. Blue, purple, and red lines represent oxygen diffusion distance $\theta$ of 0 (no oxygen), 1 (low oxygen), and ∞ (high oxygen), respectively. Relative to anaerobic or highly aerobic conditions, intermediate oxygen availability strongly suppresses the evolution of large size (shown as a function of O₂ diffusion depth in **c**, and as a function of environmental pO₂ in **d**). In both **c** and **d**, dashed lines denote $k = 0.1$, while solid lines denote $k = 0.00001$. With smaller values of $k$, organisms need to be larger to obtain the same survival benefit of increased size. In **d**, black, green, and red lines model the rate of O₂ consumption, $\varphi = 46\ mg\ s^{-1}\ l^{-1}$, while the blue line has $\varphi = 4.6\ mg\ s^{-1}\ l^{-1}$. The black and blue lines model the rate of O₂ diffusion, $D_e = 1.12 \times 10^{-5}\ cm^2\ s^{-1}$, green line $D_e = 2.24 \times 10^{-5}\ cm^2\ s^{-1}$, red line $D_e = 5.56 \times 10^{-5}\ cm^2\ s^{-1}$. In all figures, solid lines denote $k = 0.00001$ and dashed lines denote $k = 0.1$. MATLAB code to generate each of these figures can be found in Supplementary Code 1.

diffusion, smaller organisms gain a fitness advantage because a larger proportion of their cells have access to oxygen, increasing their per-cell metabolic returns. As a result, the evolution of large size can be highly constrained under low-O₂ conditions, but is unconstrained by this selection when O₂ is either absent or in great abundance.

Our results show striking parallels with historical trends. While fossils of putative macroscopic multicellular organisms have been identified during low-O₂ periods of Earth's history[66,67], it was not until atmospheric oxygen rose to relatively high levels in the late Proterozoic that we see unequivocal evidence for the rapid rise and ecological expansion of millimeter-to-centimeter scale multicellularity[11,17–19,26,68]. Stem group eukaryotes arose ~1.8 billion years ago, after the Great Oxidation Event (2.4 Ga), but did not evolve macroscopic multicellularity for over a billion years. During this so-called "boring billion"[49] oxygen was present in surface marine environments but at relatively low levels[12,69]—conditions that our work suggests could have strongly constrained the evolution of large size. Recent work has shown that evolutionary transitions, both from unicellularity to multicellularity[54,70,71] and from microscopic to macroscopic multicellularity[7], are not necessarily highly constrained by innate biological limitations. Indeed, snowflake yeast evolve a 4.5-fold increase in maximum cluster size over just 145 days (radius of the 95th percentile ancestor = 39.8 μm, radius of the 95th percentile day-145 supplemental oxygen = 180 μm), a miniscule interval with respect to geologic time. Ongoing research into the state and dynamics of early Earth surface environments, together with an inclusive consideration of evolutionary trade-offs, is thus a particularly exciting avenue for discovering how environmental constraints may have acted to shape the evolution of complex multicellular life[13,25–28,31,72,73].

The evolution of macroscopic multicellularity is a complex topic with no single causal agent[26,74]. Several macroevolutionary trends are becoming apparent, however. The evolution of large, physically-robust bodies generally requires cellular specialization[3,4] in addition to a permissive oxygen environment, and this may evolve through long-term evolutionary feedbacks[26]. For example, cell types capable of transporting oxygen and nutrients within organisms may have evolved through feedbacks with physical mechanisms allowing for large organismal size, leading to dramatic increases in organismal complexity[5]. At least within eukaryotes, aerobic metabolism is important not just from an energetic perspective, but also as a mechanism for generating molecules (e.g., reactive oxygen species) that play a key role in development and intercellular communication[75]. Developing a conceptual synthesis for the origin and subsequent evolution of complex multicellular life will thus require a transdisciplinary effort, including novel first-principles approaches that will allow us to understand, not simply characterize, historical patterns of macroevolutionary change.

## Methods

**Snowflake yeast as a model of undifferentiated multicellularity.** We used snowflake yeast to examine the relationship between oxygen availability and the evolution of multicellular size[50,52,54]. *Saccharomyces cerevisiae* is a yeast that typically grows as single cells. In nature, size-based selection may result from a number of conditions, including predation, resistance to abiotic stresses, or for the benefits of cooperative metabolism[70,76–78]. Here, we favor larger size through daily selection for rapid settling through liquid media. Larger clusters settle through liquid media more rapidly than smaller clusters, allowing us to quickly select for

large size across large numbers (~400,000 in the ancestral strain) of individual snowflake clusters. We stress, however, that settling selection is not meant to directly recapitulate any particular environmental selection for multicellularity, but rather provides a powerful means for selecting directly on organism size—a step at the heart of any evolutionary transition to multicellularity.

**Strain construction.** All of our experiments begin with an isogenic clone of a diploid yeast, strain Y55, engineered to form "snowflake" clusters by deletion of both copies of the reading frame encoding the *ACE2* transcription factor[52] (*ace2Δ::KANMX/ace2Δ::KANMX*, or *ace2Δ*, all primer sequences are listed in Supplementary Data 2). This basic *ace2Δ* snowflake yeast strain is naturally mixotrophic, capable of both fermentation and aerobic respiration. From this basic snowflake yeast, we generated metabolically anaerobic yeast by selecting a randomly generated petite mutant (i.e., carrying deletions on its mitochondrial DNA that render it nonfunctional). This anaerobic yeast phenotype strictly ferments its carbon and cannot consume $O_2$ for its energy metabolism. We confirmed the fact that this strain cannot respire by showing it was incapable of growth on glycerol, a non-fermentable carbon source. We also confirmed that it does not consume $O_2$ during growth with direct $O_2$ measurements using $O_2$ optodes. For the third and final strain, which is strictly aerobic, we simply grew snowflake yeast in media where their primary carbon source (glucose) was replaced with equal molar concentration of glycerol, which cannot be fermented and can only be metabolized via aerobic respiration.

To generate a large snowflake yeast genotype for the competition assays (relative fitness analysis, Fig. 3), we deleted two genes, *GIN4* and *ARP5*, in the *ace2Δ* background. These two genes were selected from a list of 36 de novo mutations from a large snowflake yeast isolate that evolved under selection for larger size (Supplementary Data 1 for a complete list of mutations). Among those 36 mutations, we screened the deletions of seven genes (i.e., *ARP5*, *GIN4*, *MEP2*, *RPA49*, *ENT4*, *MCM6*, and *MLP2*) for their potentially positive effect on cluster size in our basic strain (*ace2Δ*). The deletions of *ARP5* and *GIN4* increased the size of the snowflake yeast clusters visibly. We brought these mutations together under the same genetic background to obtain even larger clusters. To do this, in diploid single strain isolates of our basic strain (*ace2Δ*), we generated two strains, one of which is heterozygous *arp5Δ::HYGNT1/ARP5*, while the other was *gin4Δ::NATNT2/GIN4*. Next, we induced sporulation and meiosis by incubating both transformants in 5 ml KAc (2% potassium acetate, 0.5 g dextrose). To collect haploid transformants, we digested tetrads in 15 μl (seven units 100 μl⁻¹) zymolyase (Zymo Research EU, Freiburg, Germany). We dissected tetrads on YEPD plates (1% yeast extract, 2% peptone, 2% dextrose, and 1.5% agar) by using a tetrad-dissection microscope platform (The SporePlay, Singer Instruments, Watchet, UK). By replica plating four-spore colonies onto either Hygromycin-B (Enzo Life Sciences) or Nourseothricin Sulfate (Gold Biotechnology Inc., U.S.)-containing plates, we obtained two viable and two inviable spores for each genotype. Through autodiploidization of germinated viable colonies, we obtained two homozygous deletion strains for each locus (*arp5Δ* or *gin4Δ*). To bring these two mutations (in addition to *ace2Δ*) together under the same genetic background, we separately induced sporulation, digested spore walls, and by mixing them in YEPD, allowed them to fertilize by outcrossing. By plating mated isolates onto media containing both drugs, we obtained our large snowflake yeast genotype with all three mutations (*ace2Δ+arp5Δ+gin4Δ*). From this triple mutant (*ace2Δ+arp5Δ+gin4Δ*) strain, which is referred to as "grande" due to its intact mitochondrial respiratory phenotype, we also selected three biological replicates of randomly produced a large "petite" mutant phenotype, which is incapable of respiration. As for the small genotype used in the competition, we simply used our basic *ace2Δ*-only strain.

To be able to measure the relative proportion of small vs large isolates in our relative fitness assays, we tagged our small strains (of both petite and grande) with a red fluorescence protein. To do that, we amplified the *pFA6a-prTEF2-dTomato-terminator_ADH1-NATMX4* construct from a pFA6a-tdTomato plasmid and inserted it into the *URA3* locus. We confirmed red fluorescent activity of Nourseothricin Sulfate resistant transformants of petite and grande snowflake yeast via fluorescent microscopy, in comparison to lack of fluorescent signal in non-transformed parental strains.

We generated the mitoloc strains that are used to test $O_2$ diffusion limitation in snowflake yeast by amplifying the *NATMX6-preSU9-yeGFP-preCOX4-mCherry* construct from the pMitoLoc plasmid (Addgene #58980) using the primer set that is listed in Supplementary Data 2. We transformed PCR products into the genome at the *LEU2* locus of grande snowflake yeast. We recovered nourseothricin sulfate resistant transformants from the plates, and confirmed the expression of the *preSU9-yeGFP-preCOX4-mCherry* construct by imaging green and red fluorescence activity at the mitochondria using a Nikon Eclipse Ti inverted microscope. Images of these strains were captured after growing them in YEP-Glycerol (1% yeast extract, 2% peptone, and 2.5% glycerol) after 7 h of growth in fresh media on the 2nd and 5th day of transfers under intermediate $O_2$ or supplemental $O_2$ conditions.

**Experimental evolution of size as a function of oxygen metabolism.** To test the evolutionary effect of pO$_2$ on organismal size, we applied selection for size (via settling selection) under three distinct $O_2$ regimens with a total of four treatment conditions: (1) Anaerobic (petite) yeast, corresponding to 0% PAL metabolically available $O_2$; (2) Strictly aerobic yeast under ~24% PAL $O_2$, (3) Strictly aerobic yeast under ~72% PAL $O_2$, and (4) mixotrophic yeast under ~25% PAL $O_2$ as a

control. For each treatment group, we evolved five replicate populations, for a total of 20 populations. Anaerobic and mixotrophic snowflake yeast were grown in YEPD (1% yeast extract, 2% peptone, 2% dextrose) and strictly aerobic yeast were grown in YEP-Glycerol (1% yeast extract, 2% peptone, 2.5% glycerol). We grew these populations in 10 ml liquid cultures in large diameter (25 mm) test tubes with rapid mixing (250 rpm) at 30 °C. For the supplemental $O_2$ treatment, we sparged humidified air into the culture tubes (utilizing the approach developed in ref. [79]) throughout the culture cycle. To apply directional selection for large size in snowflake yeast, we took stationary-phase populations that had grown for 24 h and transferred 1.5 ml (approximately 400,000 clusters) of each culture into 2 ml centrifuge tubes and let them settle on bench top for 4 min. Next, we discarded top 1450 μl of these samples and transferred the bottom 50 μl into 10 ml of fresh media for the next round of growth. In total, we applied 145 consecutive rounds of growth and settling selection, or ~812 snowflake yeast generations. We measured the dilution factor used to calculate[80] the approximate number of generations snowflake yeast undergoes with a Sysmex Cyflow Cube 6 flow cytometer, measuring the daily fold change in cluster number for all 20 populations at after 0, 50, 100, and 145 days of evolution. This allows us to account for changes in the daily dilution factor (and thus number of generations per day), that occurred over our experiment. We measured average $O_2$ in each treatment group for all populations by using a fiber-optic $O_2$ optode that provides real-time $O_2$ monitoring (FireStingO$_2$, PyroScience, GmbH, Germany) (raw data for average pO$_2$ measured for each population can be found in the Source Data file). Finally, we also stored weekly frozen glycerol stocks of each population at −80 °C.

**Fitness assays on genetically-engineered small and large clusters.** To examine the relative fitness of genetically engineered "large" clusters vs. "small" clusters, we recapitulated the same treatments that were used in our evolution experiment. Those conditions are as follows: (1) competition of two anaerobic (i.e., petite) phenotypes (0% PAL metabolically available $O_2$), (2) competition of two mixotrophic phenotypes (~25% PAL $O_2$), (3) competition of two strictly aerobic phenotypes (~25% PAL $O_2$), and (4) competition of two strictly aerobic phenotypes with supplemental oxygen (~75–80% PAL $O_2$). To start the competition assays, we first grew four strains in monocultures overnight. These strains were: (1) "small petite" (*ura3::dTomato/URA3+ace2Δ*), (2) "large petite" (*ace2Δ+arp5Δ+gin4Δ*), (3) "small grande" (*ura3::dTomato/URA3 +ace2Δ*), (4) "large grande" (*ace2Δ+arp5Δ+gin4Δ*). From these monocultures, we prepared mixtures of small and large phenotypes, separately for petite and grande snowflake yeast. We measured the starting proportions of small vs. large yeast via fluorescent microscopy. Next, we inoculated 100 μl of each mixture into 10 ml of appropriate fresh media for the first day of co-culturing. In total, we applied three days of growth and two rounds of settling selection. To minimize the potential for evolution to occur within these fitness assays, we increased the strength of settling selection by decreasing the duration of settling selection from 4 to 3 min. We measured the final proportions of each phenotype via fluorescent microscopy and then calculated the average daily change in frequency of large vs. small yeast daily in each competition. When calculating the frequency of each genotype, we counted every snowflake yeast cluster as one individual. Because group size distributions were stable (see "Measuring multicellular size" below), changes in the numbers of groups should also closely reflect changes in the number of cells and alleles. We measured relative fitness by calculating the average change in frequency, rather than the more common method of calculating the ratio of Malthusian growth parameters, because our analysis pipeline used fluorescent microscopy to differentiate strains, allowing us to easily calculate relative frequencies of each strain, but not their overall abundance (raw data in the Source Data file).

**Measuring multicellular size.** We measured the size of snowflake yeast within all 20 populations after 50, 100, and 145 days of evolution. Prior to measuring their size, we did an experiment to determine the stability of the size distribution. We generated a quantile–quantile plot, regressing cluster size at each percentile from clusters grown for 12 h against the comparable percentiles from 6 h of growth. This relationship was linear with a slope near 1, $y = 0.95x + 0.33$, $r^2 = 0.9992$. Therefore, we decided to measure the size during this period in which the size distribution was stable, after 10–12 h of daily growth. To measure size, we inoculated three replicates of each population in liquid media for overnight growth at 30 °C. We transferred 100 μl of each culture into tubes with 10 ml of fresh media and grew them for 10–12 h by shaking them at 250 rpm. To measure the multicellular size distributions of each sample, we used the Multisizer-4e Coulter particle sizing and counting device (Multisizer 4 Software, Beckman Coulter, Inc.), which allows us to measure particles that are between 5.6 and 224 μm (aperture diameter = 280 μm). For each sample, we collected size data from 2000 to 4000 snowflake yeast clusters (Source Data file). Between each sample run, we ran water and measured the noise produced in cell-free solution. This noise was relatively low (<5% of sample counts), and were of small size (<20 μm). We have included representative confocal (2×2 frames, ×200 magnification) images of each evolving population after 145 transfers in the supplement, imaged on a Nikon A1R confocal microscope. These images are uncropped and capture the typical size range of each population, and we find it helpful for putting the quantitative data from the Coulter counter in perspective.

To measure the size of genetically engineered small and large snowflake yeast, we used a microscopy-based size measurement method. We grew single (*ace2Δ*)

and triple ($ace2\Delta+arp5\Delta+gin4\Delta$) mutant snowflake (petite and grande) yeast as described for above in the Multisizer experiment. We pipetted each of the four strains onto well microscope slides with three replicates, and then captured 25 congruent fields of view using a Nikon Eclipse Ti inverted microscope at 100-fold magnification. We used ImageJ-Fiji to detect and measure cluster size (Version 2.1.0). In total, we measured the size of 10,763 snowflake yeast clusters (Source Data file).

We used microscopy to measure the maximum (not just average) change in size between our ancestor and high-O₂ evolution lines, as rare but very large groups tend to be poorly resolved on the Coulter Counter. For this, we measured the two-dimensional cross-sectional area of each cluster. We grew batch cultures of ancestral and evolved populations in 10 ml YEPG for 12 h, pipetted cultures onto well-slides (1 mm in depth), and took 25 stitched images at 100-fold magnification using a Nikon Eclipse Ti inverted microscope. In total, we measured 669 clusters for the ancestral (strictly aerobic) population and 1519 clusters for the five evolved (supplemental O₂) populations (Source Data file). Finally, we calculated the mean size of the 95th percentile for the ancestral and evolved lines, using this as our estimate of the maximum size strain is capable of obtaining.

**Measuring cellular aspect ratio**. We inoculated 23 yeast populations (3 ancestors and 20 evolved populations after 145 transfers) for overnight growth at 30 °C. We then transferred 100 μl of each culture into tubes with 10 mL of fresh media, and grew them for 12 h by shaking them at 250 rpm in a 30 °C incubator. After washing 500 μl of each culture and resuspending in 500 μl calcofluor-white (5 μl of 10 μg ml⁻¹ stock solution), we fluorescently labeled cell walls by incubating them in the dark for 30 min. We washed and imaged the samples using a Nikon Eclipse Ti microscope. We measured the aspect ratio of individual cells within snowflake yeast clusters on ImageJ-Fiji[81]. In total, we collected aspect ratio data for 9553 cells (a mean of 434 cells per population).

**DNA extraction, genome sequencing, and identification of de novo mutations in a large-sized snowflake yeast isolate**. In order to identify a list of candidate mutations for the purposes of genetically engineering a large cluster for use in our fitness assays, we sequenced the whole genome of a large-sized snowflake yeast isolate. We extracted genomic DNA from this evolved isolate and its ancestor, using a commercially available kit (Amresco, Inc. VWR, USA). We prepared the genomic DNA library using NEBNext Ultra DNA Library Prep Kit for Illumina (New England Biolabs, Inc). We sequenced the genomes of the ancestral and evolved strain using the HiSeq 2500 platform (Illumina, Inc). To identify mutations, we first filtered out low quality reads using Trimmomatic (v0.39)[82]. We aligned FASTQ files to the yeast reference genome using BWA-MEM[83]. We called SNPs using GATK4 HaplotypeCaller (v4.0.3.0)[84], filtered low quality variants calls VCFTOOLS[85]. By making a pairwise comparison between the VCF files of the evolved and the ancestral strain (bcftools-isec, v1.10)[86], we extracted de novo variants. This final VCF file was annotated by using SnpEff (v4.3T)[87]. The list of all mutations identified in the evolved, large snowflake yeast isolate can be found in Supplementary Data 1.

**Statistical analysis**. Linear regression was performed using the "scipy-stats" package (1.6.0) in Python[88]. One-way ANOVA and following multiple comparison tests were performed using Prism GraphPad version 8.4.2 for Mac OS.

**Mathematical modeling**. We implemented the mathematical model described in the text in MATLAB. To generate the plots in Fig. 4, we iterated over values of $\theta$ from 1 to 100, computing the fitness of organisms with volume $s$ ranging from 1 to 100,000. To compute the fitness for a given value of $\theta$ and $s$, we solved the equation $ds/dt = \lambda_a(s)s$ to determine the doubling time $\tau(O,s)$, which in turn allowed us to directly evaluate the fitness of organisms of each size under different oxygen conditions using Eq. 1. All four plots in Fig. 4 can be recapitulated via the MATLAB script appended as Supplementary Code 1.

**Reporting summary**. Further information on research design is available in the Nature Research Reporting Summary linked to this article.

## Data availability

All strains and microscopy images are available upon request. Whole genome sequencing reads of the large-sized snowflake yeast isolate (i.e., with mutations on ARP5 and GIN4) have been deposited at the Sequence Read Archive (SRA): PRJNA719855. Source data are provided with this paper.

## Code availability

Matlab code used to generate Fig. 4a–d is included in Supplementary Code 1.

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

## Acknowledgements

We are grateful to Troy von Beck who helped us setting up the air-sparging system for our supplemental-oxygen treatment experiment. We thank Thomas C. Day for helping us with the confocal microscopy. Andrew P. Shaw trained us to use the Multisizer-4e instrument located at the Petit Institute core facilities at Georgia-Tech. We would like to thank Anthony Burnetti whose own work inspired us to genetically engineer the large snowflake yeast phenotype. The narrative structure of the manuscript was improved by feedback from Pedro Márquez-Zacarías and Peter Conlin. This work was supported by NSF DEB-1845363 to W.C.R, NSF grant no. IOS-1656549 to W.C.R., NSF grant no. IOS-1656849 to E.L., and a Packard Foundation Fellowship for Science and Engineering to W. C.R. C.T.R. and W.C.R. acknowledge funding from the NASA Astrobiology Institute.

## Author contributions

G.O.B., W.C.R. and C.T.R. conceived of the project. G.O.B. planned, performed, and analyzed the experiments. R.P. assisted with the fitness assays. E.L. and W.C.R. performed the modeling. G.O.B. and W.C.R. wrote the first draft of the paper, and all the authors contributed to the revisions.

## Competing interests

The authors declare no competing interests.

**Additional information**

