## [Peer Review File · Nature Communications]

Reviewers' Comments:

Reviewer #1:

Remarks to the Author:

Nature Communications Review: Oxygen suppression of macroscopic multicellularity

This paper seeks to test the role of oxygen availability and concentration on the evolution of "simple" multicellularity. Specifically, they seek to test the Oxygen Concentration Hypothesis using experimental evolution with snowflake yeast in different oxygen treatments. The authors follow up their experimental results with modeling results that recapitulate their findings. This was an interesting experiment with interesting findings.

I have some specific comments below, but the broad concern/question I have relates to the generality of the yeast system to address this question. The argument that the relationship is not monotonic rests on treatments that allow the yeast to ferment anaerobically, something yeast have evolved for millions of years to do. (It is of course this trait that has led them to have the outsized role in human history that they do!) It is therefore not surprising that the anaerobic treatment was able to adapt so readily in the no-oxygen treatment. How common is this kind of very efficient metabolic-switching thought to be in early eukaryotes? How many of these early organisms could grow readily and efficiently in anaerobic and aerobic conditions, and easily switch when an anaerobic core was generated? I think the authors need to provide more background of other microbes/groups to make the results of this paper convincing as a general finding. To argue that oxygen may have in fact suppressed the evolution of larger size of microbial communities, there should be a fuller discussion of more than just Crabtree specialized yeast.

A separate question is whether the OCH is meant to consider anaerobes at all. From the manuscript: "The OCH posits that, in organisms that lack a circulatory system, the ability for oxygen to diffuse into an organism places limits on organism size. The OCH thus predicts that increasing atmospheric pO₂ should generally increase the depth to which O₂ can diffuse, monotonically increasing the maximum size that can be attained before diffusive O₂ limitation impedes growth." In this more specific way, then the results presented here do in fact support the OCH. When oxygen is required for metabolism, there is a monotonic relationship.

For these reasons, I think that a resubmission should clarify: (1) if the argument they are making is that the OCH is not supported because the possibility of switching between anaerobic and aerobic metabolize subverts the pressure on oxygen concentration, and (2) whether or not there is evidence that efficient metabolic switching was common in early eukaryotes. And of course all of this (the question, the experiment, and the model) rests on the assumption that bigger is better, which may be reasonable, but it is a big assumption.

Separate from the issue of the generality of the results, I think that this is a very nice yeast experiment whose modeling does an impressive job of recapitulating the experimental work. It is very cool to directly show that snowflake size can be limited by oxygen on a non-fermentable food source and that anaerobic snowflakes can get just as big as aerobic ones. [Although related to the broad comment above, it must be noted that all the parameters and assumptions in the model are *S. cerevisiae*-specific (e.g., respiration yields 6.8 times as much growth as fermentation).]

Minor comments about the experimental work:

(1) According to the *Saccharomyces* Genome Database, *arp5* null mutants have decreased fermentation and competitive fitness, and *gin4* null mutants have decreased competitive fitness and slower growth rates. How might this affect the broader conclusion of the engineered competition? It may not, but it is worth noting in the manuscript that there are other phenotypic effects of these genetic alterations.

(2) With regard to the competitions, the manuscript never quite states what is being counted to estimate fitness. It is noted as "the change in genotype frequency" in the Results and change in "the relative frequencies of each strain" in the Methods. Is it per cell? Per cluster? Do you start the competition with the same number of cells, clusters, or liquid culture from each strain? Please specify.

(3) The last sentence of the Discussion is a bit heavy-handed and it is unclear what the difference between "understand" and "explain" is supposed to be. If one understands something, one should be able to explain it. Perhaps the authors mean characterize vs. understand/explain.

Reviewer #2:

Remarks to the Author:

In this article, the authors examine the effect of oxygen on the evolution of multicellular size using a combination of experimental evolution, synthetic biology, and mathematical modelling.

I will be focussing on the modelling part of the article, which is the one I was asked to assess.

The mathematical model considered is a very simple evolutionary model which consists of an equation to calculate the fitness of an individual of a given size at a given time under the environmental conditions dictated by the availability of oxygen. Despite its simplicity, the model appears to serve its purpose of contextualising the experimental results obtained and providing general insight into the role of oxygen in the evolution of organismal size. However, there are a number of points that need to be addressed.

Major comments.

1) On lines 207-208 it is stated that $\lambda_a(s)$ is the average cell growth rate for an organism of size s . However, on lines 214-215 it is given a definition of λ_a as a function of p_f (i.e. the proportion of anaerobic cells within an organism). For these two definitions to be consistent, there should be a one-to-one relationship between s and p_f . How are p_f and s related?

2) For each panel of Figure 3, a detailed description of how the displayed results were obtained needs to be included.

3) The results in Figure 3 refer to the case where $k=0.001$. How was this value of the parameter k chosen? How does the value of k affect the results obtained?

4) The results displayed in Figure 3b refer to the case where $t=1$. Why is it so? What does $t=1$ mean from a physical point of view? Since all parameter values are given in units of seconds, one is led to think that $t=1$ corresponds to 1 second, which does not appear to be a physically relevant time scale.

5) In the caption of Figure 3, the values of a parameter named k_o are given. However, this parameter is not mentioned in the description of the model. Is k_o meant to be the parameter named ϕ in the model description?

Minor comments

1) Line 213: "is the weighted"  "as the weighted"

2) Line 232: " $\text{cm}^2/\text{s}^{-1}$ "  " $\text{cm}^2 \text{s}^{-1}$ "

3) Line 247: " L^{-1} "  " l^{-1} "

Reviewer #3:

Remarks to the Author:

In this manuscript, Ozan Bozdog et al describe an experimental evolution experiment of 'snowflake yeasts' under different oxygen conditions to test whether oxygen limitation is a constraint on the size of diffusion-limited multicellular organisms. Overall, the concept is interesting, but I believe there is a substantial lack of experimental support for their main premise and for the modeling section of the paper. I have several major concerns that the authors should address:

1) The paper makes at least one fundamental assumption that has not been directly tested in the context of these snowflake yeast, namely that increased cluster size leads to oxygen limitation of the cells within the cluster. However, cells lacking *ace2* are not organoids and clusters may have clear 'holes' allowing free diffusion of materials within the cluster and it is not clear if the cells on the outside actually consume oxygen before it reaches the center of the cluster, especially since their cells are in well-mixed tubes. The authors need to show that oxygen is indeed not reaching the center of the clusters for the whole manuscript to be valid (presumably using some sort of oxygen biosensor).
2) The bulk of the experiments consists of pairs of comparisons, rather than 4 data points as often plotted in the figures. The experiments consist of petite vs grande in YPD, and grandes in YPGly at different O₂ concentrations.

Why are the grande cells smaller than petite cells in YPD after evolution? Presumably here the cells are under very little oxygen demand as they are mostly fermenting. Have the authors performed a similar experiment but with grande cells in YPD at 70% PAL to show that indeed grande cells are oxygen starved in YPD and this imposes growth limitation?

Here the authors show this indirectly by adding mutations to grande cells that make them larger (Fig 2b). But this reconstruction shows the same fitness defect as grande cells in YPGly which has presumably full dependence on oxygen. Assuming that cluster sizes are not affected by YPD vs YPGly, this seems highly unusual.

Further, have the authors shown that oxygen has no fitness effect for these mutations on the petites?
3) The modeling section was interesting, however I was a bit confused by some of the choices that the authors choose to do. What is the modeling justification for $p(s) = 1 - 0.5 \exp(-ks)$ as an approximation of their selection regime and is the 'k' they chose a free parameter to make data fit or is this something that can be empirically determined?

It seems the experiment the authors perform has two major components that determine the fitness of a cluster size. First is the growth rate of the cluster, which translates to fitness through usual logit function. Second is the probability of avoiding the aspiration after some amount of time of settling. This settling force is gravity and viscous drag, which can be approximated to have a velocity of the square of the radius of the cluster. Thus, size should increase the probability of being sampled at the next generation in a growth independent process.

However I am unable to recognize these processes in the formula. It is possible that this ends up as what the authors have written (or that their formula is an approximation of this), but it would be good if they expanded on the modeling section in the methods.

4) In figure 3a, this seems to be a modeling decision. But is it not possible to obtain this data directly experimentally? I'm not convinced that the curve looks like this. Since yeast prefers to ferment even with oxygen, it seems that these curves might not be a good representation of the experimental model.

5) The section describing how the authors calculate relative fitness needs more clarification. The text makes it seem as though the authors are counting the proportion of alleles (individual cells - line 173) vs the proportion of clusters. I cannot find any experiments showing that the authors can count accurately the number of cells from the clusters.

6) What is the physiological basis for the mutations they introduce into their cells that makes clusters larger? In Wei Yao et al (2016), the authors show that loss of *arp5* may shift the balance of cells towards respiratory behaviors. Is it possible that cell-size is a proxy for growth rate and that *arp5* makes the cells grow faster?

7) It seems that cluster size must be convolved with growth rate, as a slow growing cell will never form larger clusters if it is not dividing fast enough. The clusters may then seed new clusters (in an unknown mechanism to me). Have the authors shown that they are measuring cluster sizes at equilibrium?

8) The settling time during the evolution was 4 minutes. The settling time during the fitness assay was 3 minutes. I'm not sure the authors have explained why.

I have a few minor concerns:

9) Do the authors have Figure 2b with single mutants of *gin4* and *arp5*?

10) The experimental reconstruction test the effect of several mutations that increase cluster size against ancestral strains. However, the authors do the fitness assay by deleting one copy of the *URA3* gene. I'm not sure if it is known that a single copy of the *URA3* gene fully complements two copies of the *URA3* gene in various oxygen conditions, especially since the pathway is tightly linked with metabolism and mitochondria. It's been well shown that even in YPD media, the gene strongly affects growth, and so the authors should verify that the selection coefficients they are measuring are not due to the lack of *URA3* by competing against knocked out *URA3* or strains expressing a different fluorophore.

11) Line 535: Glycerol should be given as a % and not as a volume if we don't know the total volume of the media prepared.

RESPONSE TO REFEREES

A note to all referees:

There is no greater gift to be given in academia than thinking hard about someone else's work and giving thoughtful and constructive feedback, and for this, we are exceptionally grateful. These are some of the most helpful reviews any of us have ever had, and the revised manuscript is considerably improved as a result. Thank you.

Referee #1:

This paper seeks to test the role of oxygen availability and concentration on the evolution of "simple" multicellularity. Specifically, they seek to test the Oxygen Concentration Hypothesis using experimental evolution with snowflake yeast in different oxygen treatments. The authors follow up their experimental results with modeling results that recapitulate their findings. This was an interesting experiment with interesting findings.

I have some specific comments below, but the broad concern/question I have relates to the generality of the yeast system to address this question. The argument that the relationship is not monotonic rests on treatments that allow the yeast to ferment anaerobically, something yeast have evolved for millions of years to do. (It is of course this trait that has led them to have the outsized role in human history that they do!) It is therefore not surprising that the anaerobic treatment was able to adapt so readily in the no-oxygen treatment. How common is this kind of very efficient metabolic-switching thought to be in early eukaryotes? How many of these early organisms could grow readily and efficiently in anaerobic and aerobic conditions, and easily switch when an anaerobic core was generated? I think the authors need to provide more background of other microbes/groups to make the results of this paper convincing as a general finding. To argue that oxygen may have in fact suppressed the evolution of larger size of microbial communities, there should a fuller discussion of more than just Crabtree specialized yeast.

A separate question is whether the OCH is meant to consider anaerobes at all. From the manuscript: "The OCH posits that, in organisms that lack a circulatory system, the ability for oxygen to diffuse into an organism places limits on organism size. The OCH thus predicts that increasing atmospheric pO_2 should generally increase the depth to which O_2 can diffuse, monotonically increasing the maximum size that can be attained before diffusive O_2 limitation impedes growth." In this more specific way, then the results presented here do in fact support the OCH. When oxygen is required for metabolism, there is a monotonic relationship.

For these reasons, I think that a resubmission should clarify: (1) if the argument they are making is that the OCH is not supported because the possibility of switching between anaerobic and aerobic metabolize subverts the pressure on oxygen concentration, and (2) whether or not there is evidence that efficient metabolic switching was common in early

eukaryotes. And of course all of this (the question, the experiment, and the model) rests on the assumption that bigger is better, which may be reasonable, but it is a big assumption.

We thank the referee for these insightful points, revising our MS in light of these comments has significantly strengthened our paper. We have organized our response here around the key two points raised in the last paragraph of the section above.

Point 1: While the OCH is often discussed in general terms, we agree with the referee that it is not a general model of oxygen-size relationships. Because it only considers obligately aerobic organisms, it cannot extend to anaerobic conditions, and thus cannot be applied to the question of what happens when the atmosphere goes from an anaerobic to microaerobic state. In the OCH, organisms simply cannot grow any larger than the diffusion depth of oxygen, a limitation that is subverted in our paper by mixotrophy, giving internal cells the ability to continue growing anaerobically. In our model, we find that oxygen suppresses size whenever the payoffs from aerobic growth are higher (even by just 0.1%) than anaerobic growth. Further, our model predicts that as oxygen increases from 0 to ~1-2% PAL, this should favour smaller and smaller sizes, which also conflicts with the OCH (which posits that more O₂ always allows for larger size). This is because in our model oxygen acts as a resource, and the benefits of becoming small to better utilize it scale with its abundance (*i.e.*, when O₂ is at very low concentrations, there's little benefit to being small and good at utilizing it). This effect is even more apparent in the current version of our modeling results where we examine a larger value of k (dashed lines denote $k=0.1$, here the fitness benefits of increased size kick in for smaller organisms than they do when k is smaller, denoted by the solid lines. If k is confusing please see our new Extended Data Figure 6, the graph is a lot easier to interpret than a verbal description.):

Point 2: The second point asks whether it makes sense to consider simple multicellular organisms that are mixotrophic. Obligate aerobic respiration, like that considered by the OCH, could only have evolved in an already oxygenated environment. Indeed, mixotrophy remains a common strategy today in oxygen minimal zones (Ginger et al. 2010; Tielens et al. 2002; Martin 2017), and phylogenetic analyses suggest that stem group eukaryotes (which evolved in the low-O₂ oceans of the proterozoic) were mixotrophic (Müller et al. 2012; Fritz-

Laylin et al. 2010; Hannaert et al. 2003; Xie et al. 2021). While we do not know if early eukaryotes could switch their metabolism quickly, the rate of metabolic switching should not qualitatively impact our conclusions. If we assume that metabolic switching is slow such that interior cells are less productive with anaerobic metabolism than they would otherwise be, then this could be modeled simply by increasing the relative growth benefit of respiration to fermentation (the λ_r parameter in our model). In our model this actually increases the extent of oxygen suppression on size, but in any case, the qualitative results of our model are extremely robust to variation in this parameter (see Extended Data Figure 7, where we plot the fittest size with $\lambda_r = 6.8$ and 1.001).

In the introduction, we have added the following paragraph, which describes the limitations of the OCH and provides context about mixotrophy:

“The OCH, however, is not a general model for oxygen-size relationships, and it cannot be applied to the question of how the transition from an anaerobic to a microaerobic world would have affected organismal size. This is because the OCH assumes that organisms are obligately aerobic, a derived metabolic strategy suited to an already well-oxygenated world. Stem-group eukaryotes, for example, which evolved in the low-O₂ oceans of the Proterozoic⁴³, were mixotrophic⁴⁴⁻⁴⁷, and mixotrophy remains common today in oxygen-minimum zones⁴⁸⁻⁵⁰. Further, the OCH has focused almost exclusively on physiological rather than evolutionary timescales, examining the immediate physiological impacts of O₂ but ignoring evolutionary feedbacks that are ultimately responsible for systematic changes in multicellular size. Oxygen is a valuable resource, both increasing the efficiency of metabolism up to 16-fold⁵¹ and stoichiometrically unlocking the metabolic potential available in non-fermentable carbon⁵². Thus, the evolution of multicellular size should be viewed through the lens of evolutionary trade-offs (*i.e.*, multicellular organisms can potentially gain a benefit by being large, but may suffer increased diffusive constraints on access to O₂ as a result, reducing the metabolic benefits of using oxygen), rather than through the strictly physiological lens of the OCH.”

We have also changed our framing throughout the MS, to make it clear that we are not testing the ‘predictions of the OCH’ (because the OCH cannot extend into anaerobic conditions), but rather, are conducting a more general investigation of the role of oxygen in the evolution of multicellular size. This is a subtle but important distinction, and we appreciate the referee’s assistance in refining our argument.

Finally, we agree with the referee that the assumption that selection favours larger size is centrally important to our results. This is a widely held assumption in studies of early multicellularity (see an elegantly written justification on p. 67 of John Tyler Bonner’s book “Why Size Matters”) for a simple reason: it needs to be true for large multicellular organisms to evolve. Under the conditions where selection acts against larger size (and this is no doubt very common), organisms should stay unicellular.

Separate from the issue of the generality of the results, I think that this is a very nice yeast experiment whose modeling does an impressive job of recapitulating the experimental work. It is very cool to directly show that snowflake size can be limited by oxygen on a non-fermentable food source and that anaerobic snowflakes can get just as big as aerobic ones. [Although related to the broad comment above, it must be noted that all the parameters and assumptions in the model are *S. cerevisiae*-specific (e.g., respiration yields 6.8 times as much growth as fermentation).]

We thank the referee for the kind words! While we chose yeast parameters in the model for ease of comparison, we did run a sensitivity analysis to see how much our results depend on these values. We see the qualitative dynamics of oxygen suppression (size is minimized when oxygen is between 0 and 1-2 % PAL) with even tiny differences in yield between respiration and fermentation. The lowest we examined was a 0.1% growth advantage for respiration—if we pushed the difference any lower numerical errors during floating point computations overwhelm the results. From the Results:

“The key qualitative result of this model—in which the transition from an anaerobic to a weakly aerobic environment strongly suppresses the evolution of large size (Figure 4c)—is quite robust. Oxygen suppresses the evolution of larger size when respiration provides as little as 0.1% more growth than fermentation (Extended Data Figure 7a). Notably, this effect is even more pronounced when we compare obligately aerobic to obligately anaerobic organisms (Extended Data Figure 8), rather than the mixotrophic organisms of the base model.”

Minor comments about the experimental work:

(1) According to the *Saccharomyces* Genome Database, *arp5* null mutants have decreased fermentation and competitive fitness, and *gin4* null mutants have decreased competitive fitness and slower growth rates. How might this affect the broader conclusion of the engineered competition? It may not, but it is worth noting in the manuscript that there are other phenotypic effects of these genetic alterations.

This is an important point. The mutants do indeed have lower competitive fitness (though this does not depend on the concentration of O₂, see below). However, we do not believe that these slower growth rates affected our interpretation of the engineered competition, as all pairwise competitions pitted the ancestor against the larger but slower-growing mutant. The benefits of greater size (higher survival rates during settling selection) more than compensate for this slower growth when oxygen is absent or abundant, but not when it is limiting.

We quantified the fitness effects of each mutant in the unicellular ancestor under serial transfer with selection for growth under both intermediate and supplemental O₂ conditions. We have added the following paragraph addressing the effect of these mutations on cellular growth rates, and explain why this does not affect our interpretation of the experiment's results [this comes directly after describing the results of the experiment]:

“Both *gin4Δ* and *arp5Δ* mutations have pleiotropic effects, reducing unicellular growth rates in a similar manner under both intermediate and high-O₂ conditions in a unicellular context. Unicellular *gin4Δ* mutants had a relative fitness of 0.898 and 0.900 under intermediate-O₂ and supplemental-O₂ conditions, respectively. Similarly, unicellular *arp5Δ* had a relative fitness of 0.937 and 0.949 under intermediate-O₂ and supplemental-O₂ conditions, respectively. This reduction in growth rates should not affect our interpretation of the engineered multicellular-strain competition experiment, as this cell-level growth cost was consistent in all pairwise competitions, and the larger engineered strains nonetheless had higher fitness under anaerobic and high-O₂ conditions (but not under intermediate-O₂).”

We also now provide biophysical context for why these mutants increase the size of snowflake yeast:

“These mutations increase group size by modifying the biophysics of snowflake yeast growth. *arp5Δ* increases cellular aspect ratio, decreasing the packing fraction within clusters and increasing their biophysical toughness⁵⁵, and *gin4Δ* increases the size of bud scars, potentially increasing the strength of cell-cell connections.”

(2) With regard to the competitions, the manuscript never quite states what is being counted to estimate fitness. It is noted as “the change in genotype frequency” in the Results and change in “the relative frequencies of each strain” in the Methods. Is it per cell? Per cluster? Do you start the competition with the same number of cells, clusters, or liquid culture from each strain? Please specify.

Good point, we measured it as the change in frequency of clusters (rather than cells) over time. This is easier for us to measure, and since cluster size distributions are stable across culture cycles for each genotype (see figure below), it is also a good proxy for the change in the number of cells of each genotype.

The quantile-quantile plots above regress the size of each percentile of a population of snowflake yeast over two intervals, with size measured via flow cytometry. Between 6-12 hours, the population size structure was stable, with the quantile-quantile plot of all size percentiles being remarkably linear with a slope near 1 ($y = 0.95x + 0.33$, $r^2 = 0.9992$). All of our size measurements in the paper were made between 10 and 12 hours of growth, during this period of size distribution stability. Thus, changes in the numbers of groups should closely mirror changes in the number of cells.

We made the following changes to the text:

Results:

“For each competition, we calculated the daily selection coefficient (proportional change in the frequency of individual clusters) of large vs. small snowflake yeast, r_w , after 24h of growth and one round of settling selection for two consecutive days of growth and size selection.”

Methods:

“When calculating the frequency of each genotype, we counted every snowflake yeast cluster as one individual. Because group size distributions were stable, changes in the numbers of groups should also closely reflect changes in the number of cells and alleles.”

(3) The last sentence of the Discussion is a bit heavy-handed and it is unclear what the difference between “understand” and “explain” is supposed to be. If one understands something, one should be able to explain it. Perhaps the authors mean characterize vs. understand/explain.

We agree and have made this change, this is indeed truer to the sentiment we were intending to convey. Thanks!

Work cited:

- Ginger, M. L., Fritz-Laylin, L. K., Fulton, C., Cande, W. Z. & Dawson, S. C. Intermediary metabolism in protists: a sequence-based view of facultative anaerobic metabolism in evolutionarily diverse eukaryotes. *Protist* **161**, 642-671 (2010).
- Tielens, A. G., Rotte, C., van Hellemond, J. J. & Martin, W. Mitochondria as we don't know them. *Trends in biochemical sciences* **27**, 564-572 (2002).
- Martin, W. Unmiraculous facultative anaerobes. *Bioessays* **39**, 1700041 (2017).
- Müller, M. *et al.* Biochemistry and evolution of anaerobic energy metabolism in eukaryotes. *Microbiol Mol Biol Rev* **76**, 444-495, doi:10.1128/membr.05024-11 (2012).
- Fritz-Laylin, L. K. *et al.* The genome of *Naegleria gruberi* illuminates early eukaryotic versatility. *Cell* **140**, 631-642 (2010).
- Hannaert, V., Bringaud, F., Opperdoes, F. R. & Michels, P. A. Evolution of energy metabolism and its compartmentation in Kinetoplastida. *Kinetoplastid biology and disease* **2**, 1-30 (2003).
- Xie, R. *et al.* Expanded Asgard archaea shed new light on the origin of eukaryotes and support a 2-domain tree of life. *bioRxiv* (2021).
- Bonner, John Tyler. *Why size matters: from bacteria to blue whales*. Princeton University Press, 2011.

Referee #2:

In this article, the authors examine the effect of oxygen on the evolution of multicellular size using a combination of experimental evolution, synthetic biology, and mathematical modelling.

I will be focussing on the modelling part of the article, which is the one I was asked to assess.

The mathematical model considered is a very simple evolutionary model which consists of an equation to calculate the fitness of an individual of a given size at a given time under the environmental conditions dictated by the availability of oxygen. Despite its simplicity, the model appears to serve its purpose of contextualising the experimental results obtained and providing general insight into the role of oxygen in the evolution of organismal size. However, there are a number of points that need to be addressed.

Major comments.

1) On lines 207-208 it is stated that $\lambda_a(s)$ is the average cell growth rate for an organism of size s . However, on lines 214-215 it is given a definition of λ_a as a function of p_f (i.e. the proportion of anaerobic cells within an organism). For these two definitions to be consistent, there should be a one-to-one relationship between s and p_f . How are p_f and s related?

Good question. There is a one-to-one relationship between s and p_f . The size variable s is supposed to represent volume. The proportion of aerobic cells in a spherical organism (p_f), $p_f = (1 - \theta/r)^3$, is a function of its radius. Thus, the link between size s and radius r would simply be $s=4/3\pi r^3$. We left the two variables distinct because the organism may have other, non-spherical, structures. Though it is clear based on your comment that we should make the connection between s and r clear in this section of the text. We have modified the text to be explicit about this connection: “we calculate the proportion of anaerobic cells within the spherical organism, $p_f = (1 - \theta/r)^3$, where r is its radius ($r = (3s/4\pi)^{1/3}$) and θ is the distance from the surface”

2) For each panel of Figure 3, a detailed description of how the displayed results were obtained needs to be included.

Good point. We have now included a section in the methods section explicitly addressing this. Perhaps more importantly, we now also provide a MATLAB file that recapitulates the graphs from the four panels of this figure. Between our description of how the model was run in the methods, the parameters of the model runs in the figure caption, and the code that generated the files, our modelling should now be as transparent and reproducible as possible.

We have now added this to the Methods:

“We implemented the mathematical model described in the text in MATLAB. To generate the plots in Figure 4, we iterated over values of θ from 1-100, computing the fitness of organisms with volume s ranging from 1-100,000. To compute the fitness for a given value of θ and s , we solved the differential equation $ds/dt = \lambda_a(s) s$ to determine the doubling time $\tau(O,s)$, which in turn allowed us to directly evaluate the fitness of organisms of each size under different oxygen conditions using equation 1. All four plots in Figure 4 can be recapitulated via the MATLAB file in Supplementary File 8.”

3) The results in Figure 3 refer to the case where $k=0.001$. How was this value of the parameter k chosen? How does the value of k affect the results obtained?

We chose $k = 0.001$ because it was an intermediate value (see figure below showing how k affects size-based selection), with selection acting strongly to increase size for organisms that were between 0.2 and 10 microns in radius:

Extended Data Figure 6. Survival probability as a function of our scaling parameter k . The value of k determines the size range over which selection rewards larger organisms. When k is large (e.g., 0.1), the selective benefits of increasing size saturate at relatively small sizes. Smaller values of k shift the dynamics of size selection towards larger size classes.

We agree with the referee that it was not clear how our choice of k affected our results, so we've now plotted all our results with two levels of k : large k ($k = 0.1$) in which selection rewards larger size even when the organisms are still small, and small k ($k = 0.00001$), where the benefits of size kick in at a larger size range. Note that the k values we now use are 100-fold larger and smaller than the value of k we used in the prior version of the MS.

Figure 4 (formerly Figure 3) now includes results for $k=0.00001$ (solid lines) and $k=0.1$, (dotted lines):

We now include variation in k in our Extended Data Figure 7, which examines variation in λ_r (the relative benefit of respiration):

Extended Data Figure 7. Examining sensitivity to λ_r and k . a) Intermediate oxygen suppresses the evolution of large size even when respiration drives only 0.1% more growth than fermentation ($\lambda_r = 1.001$, $\lambda_f = 1$). Larger values of k connote that selection acts to reward larger organisms at a smaller absolute size (see Extended Data Figure 6). b) This plot is the same as for (a), but with $\lambda_r = 6.8$. The central qualitative result of our model, oxygenic suppression of size, was robust to wide variation in both the benefits of respiration (determined by λ_r) and size range over which selection acts on to favour larger organisms (determined by k).

We even incorporated variation in k into Extended Data Figure 8, which examines the effects of obligate aerobic metabolism relative to mixotrophy.

The new presentation of the model now provides context for how variation in size-based selection (as determined by k) affects quantitative dynamics of our system. Importantly, we now show that our conclusions are robust to wide (10,000-fold) variation in k .

4) The results displayed in Figure 3b refer to the case where $t=1$. Why is it so? What does $t=1$ mean from a physical point of view? Since all parameter values are given in units of seconds, one is led to think that $t=1$ corresponds to 1 second, which does not appear to be a physically relevant time scale.

Yes, this is a good question. The t value is incorporated into the fitness equation, equation 1. We can rewrite the equation as follows:

$$w(s, t) = (2p(s))^{t/\tau(O,s)} = ((2p(s))^{1/\tau(O,s)})^t = \bar{w}(s)^t$$

Where \bar{w} is the mean geometric fitness per time unit. Thus to find the total fitness we can simply raise it to the power t . In this formulation, if one organism has a higher mean geometric fitness than another then regardless of the t value, its final fitness ($w(s,t)$) will be higher. So we chose $t=1$ for simplicity but it should not mean any loss in generality. The units of time that make the most sense will depend on the units of tau which is the doubling time of the organism.

5) In the caption of Figure 3, the values of a parameter named k_o are given. However, this parameter is not mentioned in the description of the model. Is k_o meant to be the parameter named phi in the model description?

Thanks for catching that typo. It is the result of a marriage of two sets of parameters that both love the letter k. We have corrected this, thanks again!

Minor comments

1) Line 213: "is the weighted"  "as the weighted"

2) Line 232: " $\text{cm}^2/\text{s}^{-1}$ "  " $\text{cm}^2 \text{s}^{-1}$ "

3) Line 247: " L^{-1} "  " l^{-1} "

We have made all of the above corrections. Thanks!

Referee #3:

In this manuscript, Ozan Bozdog et al describe an experimental evolution experiment of ‘snowflake yeasts’ under different oxygen conditions to test whether oxygen limitation is a constraint on the size of diffusion-limited multicellular organisms. Overall, the concept is interesting, but I believe there is a substantial lack of experimental support for their main premise and for the modeling section of the paper. I have several major concerns that the authors should address:

1) The paper makes at least one fundamental assumption that has not been directly tested in the context of these snowflake yeast, namely that increased cluster size leads to oxygen limitation of the cells within the cluster. However, cells lacking *ace2* are not organoids and clusters may have clear ‘holes’ allowing free diffusion of materials within the cluster and it is not clear if the cells on the outside actually consume oxygen before it reaches the center of the cluster, especially since their cells are in well-mixed tubes. The authors need to show that

oxygen is indeed not reaching the center of the clusters for the whole manuscript to be valid (presumably using some sort of oxygen biosensor).

This is an excellent suggestion. We agree that it is important to confirm that snowflake yeast are O₂ limited under the culture conditions that we use in this MS. We took a biomarker approach, as suggested by the referee, using the pMitoLoc (Vowinckel et al. 2015) system to examine aerobic respiration as a function of cellular location. pMitoLoc drives the expression of two fluorophores, a stable preSU9-GFP that localizes to all mitochondria, and a preCOX4-mCherry that localizes to mitochondria in a membrane-potential dependent manner (*i.e.*, mitochondria with an active proton gradient). This allows us to label respiring mitochondria in red, and counterstain all mitochondria in green. Thus, cells with sufficient access to oxygen to respire should have orange mitochondria (containing both red and green stains), while those that are unable to respire oxygen should just contain green mitochondria.

We find strong gradients of aerobic respiration within our snowflake yeast clusters. The outer ~10-30 μm typically shows highly aerobic behavior (see image below), with the remainder of the cluster interior being effectively anaerobic. Under intermediate O₂ just 29% of cells contained respiring mitochondria on average; this increased to 56% with supplemental oxygen (Mann-Whitney $U = 262$, $n_1 = 29$, $n_2 = 37$, $p=0.0003$ two-tailed).

We have incorporated this data, and a description of it, into Figure 1 and the results section.

Specifically, we have added the following text:

“Snowflake yeast grow as approximately spherical clusters of densely packed cells, but little is known about the extent to which interior cells are oxygen limited- a constraint that is necessary for their use as a model system of diffusion-limited multicellularity. We examined the diffusion depth of oxygen within snowflake yeast by genomically integrating the MitoLoc construct (*preSU9-GFP + preCOX4-mCherry*)⁵⁷ into our ancestral snowflake yeast strain, allowing us to visualize mitochondrial activity throughout the cluster. The preSU9-

GFP marker localizes to the F0-ATPase subunit-9 independent of aerobic respiration, whereas preCOX4-mCherry localizes to the mitochondria only when there is an active proton gradient in the organelle. Only cells near the surface were capable of respiration, while the rest of the cluster interior is effectively anaerobic (Figure 1a). Furthermore, while an average of 29.35% of cells/cluster showed aerobic activity under standard batch culture conditions, this increased to 56% with oxygen supplementation (Figure 1b; Mann-Whitney $U = 262$, $n_1 = 29$, $n_2 = 37$, $p=0.0003$, two-tailed). Despite their branched growth form, respiration in snowflake yeast is strongly diffusion limited.”

2) The bulk of the experiments consists of pairs of comparisons, rather than 4 data points as often plotted in the figures. The experiments consist of petite vs grande in YPD, and grandes in YPGly at different O₂ concentrations.

Why are the grande cells smaller than petite cells in YPD after evolution? Presumably here the cells are under very little oxygen demand as they are mostly fermenting. Have the authors performed a similar experiment but with grande cells in YPD at 70% PAL to show that indeed grande cells are oxygen starved in YPD and this imposes growth limitation?

This is an interesting question. While we do not know why petites have evolved larger cells than grandes, our best guess is that cell size is evolving primarily due to its effects on overall cluster size. Larger cells, but primarily longer cells (higher length relative to width) are the primary way that snowflake yeast evolve to be larger (Jacobeen et al., Nature Physics 2018 and Jacobeen et al., Physical Review E, 2018). And in our experiments here, we see a linear correlation between cellular aspect ratio and cluster size (Figure 2d). In our ongoing experimental evolution experiments with petites, we have seen them evolve aspect ratios in excess of 4 after 600 transfers—they look more like bacilliform bacterium than yeast! The converse is likely true as well—if selection favors smaller multicellular size in grande yeast, then the easiest way to accomplish this is by evolving smaller, rounder cells.

Here the authors show this indirectly by adding mutations to grande cells that make them larger (Fig 2b). But this reconstruction shows the same fitness defect as grande cells in YPGly which has presumably full dependence on oxygen. Assuming that cluster sizes are not affected by YPD vs YPGly, this seems highly unusual.

Good observation. We were surprised by this too, until we started measuring oxygen within our cultures directly. We saw extensive oxygen use in YPD as well as YPGly in our experiments (both become oxygen limited at stationary phase), so this may explain why they behave similarly.

Further, have the authors shown that oxygen has no fitness effect for these mutations on the petites?

Good question, we had not done that, but it is a good idea. We re-ran our fitness assay for small vs. large petite synthetic pairs (differing in $\Delta arp5$ and $\Delta gin4$) under supplemental oxygen. The large petite strain still had a significant fitness advantage, possessing an average $r_w = 19.4\%$ ($n=3$), which is similar to the outcome of this experiment in intermediate oxygen. Also, we examined the fitness effects of an $arp5\Delta$ or $gin4\Delta$ mutation under intermediate and supplemental oxygen in the grande, and found no effect of oxygen environment on the fitness effects of these mutations (see our response to point 6 below for a detailed explanation of this experiment).

3) The modeling section was interesting, however I was a bit confused by some of the choices that the authors chose to make. What is the modeling justification for $p(s) = 1 - 0.5 \exp(-ks)$ as an approximation of their selection regime and is the 'k' they chose a free parameter to make data fit or is this something that can be empirically determined?

It seems the experiment the authors perform has two major components that determine the fitness of a cluster size. First is the growth rate of the cluster, which translates to fitness through usual logit function. Second is the probability of avoiding the aspiration after some amount of time of settling. This settling force is gravity and viscous drag, which can be approximated to have a velocity of the square of the radius of the cluster. Thus, size should increase the probability of being sampled at the next generation in a growth independent process.

However I am unable to recognize these processes in the formula. It is possible that this ends up as what the authors have written (or that their formula is an approximation of this), but it would be good if they expanded on the modeling section in the methods.

This is a good question, and the fault is ours for not clarifying our modeling approach in the manuscript. Our model is not intended to recapitulate our experimental set up, but rather, examine the logic of how oxygen would affect selection on organismal size in a more general manner. We have now added a sentence to the beginning of the modeling section stating this explicitly:

“This model is not intended to directly recapitulate our experimental results above, but instead, examine how oxygen affects the evolution of multicellular size more generally.”

And have changed the name of the modeling subsection from “Evolutionary model” to “General evolutionary model of O₂-size relationships.”

The function you refer to above is a size-based fitness function. k is not a free-parameter chosen to make the outcome fit a preconceived notion of how it should look. k is just a scaling parameter that allows us to vary the size range over which selection acts strongly on size. You can now see this in a new supplementary figure, Extended Data Figure 6.

Extended Data Figure 6. Survival probability as a function of our scaling parameter k . The value of k determines the size range over which selection rewards larger organisms. When k is large (e.g., 0.1), the selective benefits of increasing size saturate at relatively small sizes. Smaller values of k shift the dynamics of size selection towards larger size classes.

We also added a new supplementary figure showing that the qualitative conclusions of our model are robust to 10,000-fold variation in k (Extended Data Figure 7b).

Extended Data Figure 7. Examining sensitivity to λ_r and k . a) Intermediate oxygen suppresses the evolution of large size even when respiration drives only 0.1% more growth than fermentation ($\lambda_r = 1.001$, $\lambda_f = 1$). Larger values of k connote that selection acts to reward larger organisms at a smaller absolute size (see Extended Data Figure 6). b) This plot is the same as for (a), but with $\lambda_r = 6.8$. The central qualitative result of our model, oxygenic suppression of size, was robust to wide variation in both the benefits of respiration (determined by λ_r) and the size range over which selection acts on to favour larger organisms (determined by k).

We chose this approach for two reasons: First, the survival benefits of size are often modeled using a saturating function similar to that of our model, because the specific mechanisms of size-based selection (e.g., settling selection, predation, cooperative extracellular metabolism, escape from environmental stressors, the ability to grow over competitors, etc.) are thought to have diminishing returns. Second, it is convenient from a mathematical perspective not to have to specify a ‘maximum size’ at which the probability of survival is 1, which we would have to do if we used a linear function.

While we could model a specific mechanism of size-based selection (*i.e.*, modeling settling selection, or trying to quantify k for a particular mechanism of size selection), we stress that our model is not intended to explain any specific biological system. Rather, we intend our model to be as simple as possible to capture the dynamics and logic of how oxygen can result

in the evolutionary suppression of multicellular size. As has often been the case in mathematical biology, we believe that a simple, general model has more explanatory power than one that is parameterized for a specific system.

4) In figure 3a, this seems to be a modeling decision. But is it not possible to obtain this data directly experimentally? I'm not convinced that the curve looks like this. Since yeast prefers to ferment even with oxygen, it seems that these curves might not be a good representation of the experimental model.

This is a great question, but as we explained above, we intend this to be a simple, general model of how oxygen affects selection on size rather than one that recapitulates what we see in our experimental model system. Hopefully our changes to the text in response to point #3 will clarify this for future readers!

5) The section describing how the authors calculate relative fitness needs more clarification. The text makes it seem as though the authors are counting the proportion of alleles (individual cells - line 173) vs the proportion of clusters. I cannot find any experiments showing that the authors can count accurately the number of cells from the clusters.

Good point. We measured the frequency of individual clusters. But, because the distribution of cluster size is relatively constant within a given genotype (see data in our response to your point 7 below), this also predicts changes in the overall frequency of cells / alleles. We have now clarified in the text:

Results:

“For each competition, we calculated the daily selection coefficient (proportional change in the frequency of individual clusters) of large vs. small snowflake yeast, r_w , after 24h of growth and one round of settling selection for two consecutive days of growth and size selection.”

Methods:

“When calculating the frequency of each genotype, we counted every snowflake yeast cluster as one individual. Because group size distributions were stable, changes in the numbers of groups should also closely reflect changes in the number of cells and alleles.”

6) What is the physiological basis for the mutations they introduce into their cells that makes clusters larger? In Wei Yao et al (2016), the authors show that loss of *arp5* may shift the balance of cells towards respiratory behaviors. Is it possible that cell-size is a proxy for growth rate and that *arp5* makes the cells grow faster?

Great question. We did an experiment to examine the effect of each mutation on the fitness of unicellular yeast under intermediate-O₂ and supplemental-O₂ conditions. We find that the deletion of both genes (*arp5Δ* and *gin4Δ*) has a fitness cost to the growth of unicellular yeast.

However, unlike the results obtained under the multicellular background (where the larger but slower-growing mutants had higher fitness under anaerobic and supplemental O₂ conditions and lower fitness under intermediate O₂ conditions), when in the unicellular background, the oxygen concentration did not have any effect on the fitness of these mutants. The relative fitness of *gin4Δ* mutants were 0.898 and 0.900 under intermediate-O₂ and supplemental-O₂ conditions, respectively. Similarly, the relative fitness of *arp5Δ* mutants were 0.937 and 0.949 under intermediate-O₂ and supplemental-O₂ conditions, respectively. Of course, we had to perform these experiments in a unicellular background so that we do not confound cell-level growth costs with slower growth caused by their larger cluster size reducing resource diffusion. These results confirm our interpretation that selection acting on the larger multicellular groups are responsible for the fitness differences we see in Figure 3b (was Figure 2b in the prior MS), not an underlying cell-level advantage of the mutations.

We now state in the text:

“Both *gin4Δ* and *arp5Δ* mutations have pleiotropic effects, reducing unicellular growth rates in a similar manner under both intermediate and high-O₂ conditions. Unicellular *gin4Δ* mutants had a relative fitness of 0.898 and 0.900 under intermediate-O₂ and supplemental-O₂ conditions, respectively. Similarly, unicellular *arp5Δ* had a relative fitness of 0.937 and 0.949 under intermediate-O₂ and supplemental-O₂ conditions, respectively. This reduction in growth rates should not affect our interpretation of the engineered multicellular-strain competition experiment, as this cell-level growth cost was consistent in all pairwise competitions, and the larger engineered strains nonetheless had higher fitness under anaerobic and high-O₂ conditions (but not under intermediate-O₂).”

We also realized that we never explained our biophysical rationale for why we think these mutations have the effect they do. We now note in the paper:

“These mutations increase group size by modifying the biophysics of snowflake yeast growth. *arp5Δ* increases cellular aspect ratio, decreasing the packing fraction within clusters and increasing their biophysical toughness⁵⁵, and *gin4Δ* increases the size of bud scars, potentially increasing the strength of cell-cell connections.”

7) It seems that cluster size must be convolved with growth rate, as a slow growing cell will never form larger clusters if it is not dividing fast enough. The clusters may then seed new clusters (in an unknown mechanism to me). Have the authors shown that they are measuring cluster sizes at equilibrium?

Excellent question. Cluster size is not convolved with growth rate because the size to which snowflake yeast grow is not determined by the speed of their growth under steady state conditions, but rather the size at which packing-induced strain causes cell-cell fracture (Jacobeen *et al*, 2018 Nature Physics). Notably, it doesn't take long for a fractured cluster to grow large enough to reproduce again- typically it only takes a division cycle or two. We empirically measured the dynamics of our cluster size distributions across their daily growth

cycle (under all metabolic conditions, and for 10- and 145-days evolved populations), and found that they reach equilibrium after 6 hours of growth (see below).

The quantile-quantile plots above regress the size of each percentile of a snowflake yeast population over two intervals via flow cytometry. Between 6-12 hours, the population size structure was stable, with the regression between initial and final sizes being quite linear with a slope near 1 ($y = 0.95x + 0.33$, $r^2 = 0.9992$). All of our size measurements were made between 10 and 12 hours after transfer, after this stabilization had occurred. As a result, changes in the numbers of groups should closely mirror changes in the number of cells.

We have now noted this in the Methods:

“We measured the size of snowflake yeast within all 20 populations after 50, 100, and 145 days of evolution. Prior to measuring size, we did an experiment to determine the stability of the size distribution. We generated a quantile-quantile plot, regressing cluster size at each percentile from clusters grown for 12 hours against the comparable percentiles from 6 hours of growth. This relationship was linear with a slope near 1, $y = 0.95x + 0.33$, $r^2 = 0.9992$. Therefore, we decided to measure the size during this period in which the size distribution was stable, after 10-12 hours of daily growth.”

Also, we realized that we did not mention how snowflake yeast grow and fracture, so we now describe this very briefly in the Introduction:

"We constructed our initial snowflake yeast by deleting the *ACE2* open reading frame in the unicellular strain Y55. This leads to incomplete separation of mother and daughter cells, resulting in the formation of multicellular clusters. Snowflake yeast possess an emergent multicellular life cycle in which clusters grow until packing strain generated by cellular division causes cell-cell fracture, giving rise to new snowflake yeast clusters^{54,55}"

8) The settling time during the evolution was 4 minutes. The settling time during the fitness assay was 3 minutes. I'm not sure the authors have explained why.

This is a great point, and we did not explain this difference in the text. We wanted to minimize the potential for evolution to occur within these fitness assays, and thus wanted to make the experiment as short as possible. After getting more experience with our system and understanding how these strains respond to selection, we decided to increase the strength of settling selection to 3 minutes from 4 minutes, allowing us to see differences in the fitness of each strain under competition as quickly as possible.

We now note this in the Methods:

"To minimize the potential for evolution to occur within these fitness assays, we increased the strength of settling selection by decreasing the duration of settling selection from four to three minutes."

I have a few minor concerns:

9) Do the authors have Figure 2b with single mutants of *gin4* and *arp5*?

We have the mutants, but we didn't use them in the fitness experiment because the single mutants weren't much larger than the ancestor. The double mutant is much larger making it ideal for a size-based competition experiment:

10) The experimental reconstruction tests the effect of several mutations that increase cluster size against ancestral strains. However, the authors do the fitness assay by deleting one copy of the *URA3* gene. I'm not sure if it is known that a single copy of the *URA3* gene fully complements two copies of the *URA3* gene in various oxygen conditions, especially since the pathway is tightly linked with metabolism and mitochondria. It's been well shown that even in YPD media, the gene strongly affects growth, and so the authors should verify that the selection coefficients they are measuring are not due to the lack of *URA3* by competing against knocked out *URA3* or strains expressing a different fluorophore.

Good question. To our knowledge *URA3* does not have haploinsufficiency in our strain, but we performed this experiment to be sure. We competed a *URA3/ura3Δ::GFP* hemizygous vs *URA3/URA3* homozygous in unicellular yeast grown in YPD. Importantly, we used an inducible GFP so that the GFP would not exert a fitness cost. The starting frequency of the *URA3/ura3Δ::GFP* genotype was 52.9%, and after three days of selection in three replicates, it was at a mean frequency of 51.91% (50.4%, 52.2%, and 53.14% in the three reps), demonstrating its selective neutrality.

11) Line 535: Glycerol should be given as a % and not as a volume if we don't know the total volume of the media prepared.

Good point, we have made this change. Thanks!

Work cited:

- Vowinckel, J., Hartl, J., Butler, R. & Ralser, M. MitoLoc: A method for the simultaneous quantification of mitochondrial network morphology and membrane potential in single cells. *Mitochondrion* **24**, 77-86 (2015).

- Jacobsen, S. *et al.* Cellular packing, mechanical stress and the evolution of multicellularity. *Nature physics* **14**, 286-290 (2018).

- Jacobsen, Shane, et al. "Geometry, packing, and evolutionary paths to increased multicellular size." *Physical Review E* 97.5 (2018): 050401.

Reviewers' Comments:

Reviewer #1:

Remarks to the Author:

The authors have sufficiently addressed all the concerns I had regarding the manuscript. In its current version, I find it an interesting and compelling set of experimental and computational results. They are now put in an appropriate framework.

Reviewer #2:

Remarks to the Author:

All my comments have been properly addressed.

Reviewer #3:

Remarks to the Author:

Ozan Bozdog et al have greatly improved the manuscript by making the presentation clearer, and by substantially providing evidence for their claims. I have no major issues with the paper as it is and can recommend the manuscript for publication. I do have some minor details that I think the authors can address to further improve the readability of the text but leave it to the editor and the author's best judgement to assess if these changes are required.

1. In the introduction, though I appreciate the discussion on the lack of models to test the OCH, the concept of 'snowflake yeasts' is introduced far too abruptly. The authors cannot expect the average reader to know what they mean by this and the loose explanation for how they are actually *S. cerevisiae* is not satisfactory. I would suggest changing the order a little bit (the 'model system' section in the results here has many features of an introductory paragraph).

2. Figure 1a is really striking. Figure 1b is supposed to be the summary data, but can the authors maybe provide a characteristic microscopy figure for both the intermediate and supplemental O₂? Maybe a good field of view of both sets, or at least one representative cluster from the medians of both intermediate and supplemental environments.

3. I wonder if the fitness results of unicellular *arp5/gin4* mutants can be placed into the figure instead of in the text but if the authors think it muddles the point too much it is also fine as is. The text however is a bit confusing and can be made clearer ("the larger engineered strains" - at first glance it's not clear that the authors are talking about the clusters here since the beginning of the paragraph was related to unicellular mutants). The text is further confusing, talking about 'relative growth rate' and 'fitness', when the authors measure 'fitness' in different ways (by the whole settling assay for clusters).

4. I'm not convinced by the explanation of 'reducing evolution' for the settling time differences (how much evolution can really occur during 1 minute of settling for 2 rounds of settling?). Maybe it's best to simply say that it was more convenient and that it qualitatively does not change the results which I'm sure would be more convincing to readers.

5. Finally, I still think that it's unjustified to place a line between petite and aerobic yeasts in their graphs for size vs oxygen diffusion (in Figure 4, between 0 and 10^{-4}). Is the step down really linear, or is it a piecewise function? It's very strange to put a disjointed x-axis with a fully joined linear relationship. Perhaps simply replacing that segment in the graph with a dashed line, or a disjointed line, would be more appropriate. The authors can choose to keep that in the actual data they collect (Figure 2) but I still feel uneasy with it.

Response to comments by Referee 3

Ozan Bozdog et al have greatly improved the manuscript by making the presentation clearer, and by substantially providing evidence for their claims. I have no major issues with the paper as it is and can recommend the manuscript for publication. I do have some minor details that I think the authors can address to further improve the readability of the text but leave it to the editor and the author's best judgement to assess if these changes are required.

Thanks! We really appreciate the referee's comments; they have significantly strengthened the paper. We agree with most of the additional comments below and have made suitable revisions.

1. In the introduction, though I appreciate the discussion on the lack of models to test the OCH, the concept of 'snowflake yeasts' is introduced far too abruptly. The authors cannot expect the average reader to know what they mean by this and the loose explanation for how they are actually *S. cerevisiae* is not satisfactory. I would suggest changing the order a little bit (the 'model system' section in the results here has many features of an introductory paragraph).

This is a great point, we agree it was a rather abrupt introduction to snowflake yeast. Rather than moving the 'Model System' section up to the introduction, we edited the introduction accordingly. Specifically, we combined the last two paragraphs of the introduction into a single paragraph, and removed reference to snowflake yeast.

Here is the beginning of that paragraph, which shows how we integrated the two paragraphs (this content used to be split into two separate paragraphs):

"No prior work has examined the relationship between oxygen availability and size over evolutionary timescales in a diffusion-limited multicellular organism, a gap that is partly due to the lack of suitable model systems. Here, we examine the effect of oxygen on multicellular size using a combination of experimental evolution, synthetic biology, and mathematical modeling, using yeast model system of undifferentiated multicellularity. First, we performed an ~800 generation selection experiment..."

2. Figure 1a is really striking. Figure 1b is supposed to be the summary data, but can the authors maybe provide a characteristic microscopy figure for both the intermediate and supplemental O₂? Maybe a good field of view of both sets, or at least one representative cluster from the medians of both intermediate and supplemental environments.

Good idea. We have provided an image of snowflake yeast with supplemental O₂ in Fig 1a, next to the intermediate O₂ image..

3. I wonder if the fitness results of unicellular *arp5/gin4* mutants can be placed into the figure instead of in the text but if the authors think it muddles the point too much it is also fine as is. The text however is a bit confusing and can be made clearer ("the larger engineered strains" -

at first glance it's not clear that the authors are talking about the clusters here since the beginning of the paragraph was related to unicellular mutants). The text is further confusing, talking about 'relative growth rate' and 'fitness', when the authors measure 'fitness' in different ways (by the whole settling assay for clusters).

Yes, we think that putting this fitness data in the figure distracts from the main point of the figure. However, we agree that the text is confusing and have modified it to make size clearly refer to the multicellular group, not cells, and revised to remove the term 'fitness', which can be difficult to interpret as it has many meanings.

The new sentence reads:

“This reduction in growth rates should not affect our interpretation of the engineered multicellular-strain competition experiment: despite this growth cost, the engineered strain that forms larger clusters still outcompeted the smaller competitor under both anaerobic and high-O₂ conditions (but not under intermediate O₂).”

4. I'm not convinced by the explanation of 'reducing evolution' for the settling time differences (how much evolution can really occur during 1 minute of settling for 2 rounds of settling?). Maybe it's best to simply say that it was more convenient and that it qualitatively does not change the results which I'm sure would be more convincing to readers.

We understand where the referee is coming from, but do not want to be untruthful regarding our intentions, even if it simplifies our explanation. In the past we have had strange results caused by rapid evolution during competition experiments, particularly when they run for a while, like 4-6 days (and initially it wasn't clear how long we would have to run these for). We're going to keep this one as is.

5. Finally, I still think that it's unjustified to place a line between petite and aerobic yeasts in their graphs for size vs oxygen diffusion (in Figure 4, between 0 and 10^{-4}). Is the step down really linear, or is it a piecewise function? It's very strange to put a disjointed x-axis with a fully joined linear relationship. Perhaps simply replacing that segment in the graph with a dashed line, or a disjointed line, would be more appropriate. The authors can choose to keep that in the actual data they collect (Figure 2) but I still feel uneasy with it.

Good point. We surveyed a number of papers in mathematical biology to see how they handled similar plotting situations. Taking inspiration from this, we have updated our plot so that, at the disjointed portion of the x axis, the plotted lines are now semi-transparent.